# Transcriptional reprogramming from innate immune functions to a pro-thrombotic signature by monocytes in COVID-19

Allison K. Maher [1], Katie L. Burnham[2], Emma M. Jones[1], Michelle M. H. Tan[3], Rocel C. Saputil[3], Laury Baillon[1], Claudia Selck[1], Nicolas Giang [1], Rafael Argüello [4], Clio Pillay[3], Emma Thorley[3], Charlotte-Eve Short[1], Rachael Quinlan [1], Wendy S. Barclay [1], Nichola Cooper[3], Graham P. Taylor [1], Emma E. Davenport[2] & Margarita Dominguez-Villar [1] ✉

Although alterations in myeloid cells have been observed in COVID-19, the specific underlying mechanisms are not completely understood. Here, we examine the function of classical CD14+ monocytes in patients with mild and moderate COVID-19 during the acute phase of infection and in healthy individuals. Monocytes from COVID-19 patients display altered expression of cell surface receptors and a dysfunctional metabolic profile that distinguish them from healthy monocytes. Secondary pathogen sensing ex vivo leads to defects in pro-inflammatory cytokine and type-I IFN production in moderate COVID-19 cases, together with defects in glycolysis. COVID-19 monocytes switch their gene expression profile from canonical innate immune to pro-thrombotic signatures and are functionally pro-thrombotic, both at baseline and following ex vivo stimulation with SARS-CoV-2. Transcriptionally, COVID-19 monocytes are characterized by enrichment of pathways involved in hemostasis, immunothrombosis, platelet aggregation and other accessory pathways to platelet activation and clot formation. These results identify a potential mechanism by which monocyte dysfunction may contribute to COVID-19 pathology.

COVID-19 is a respiratory tract infection caused by severe acute respiratory syndrome coronavirus 2 (SARS-CoV-2). Although the majority of infections in unvaccinated individuals are mild or asymptomatic, 15% of patients develop moderate to severe disease requiring hospitalisation, and 5% develop critical disease with life-threatening pneumonia, acute respiratory distress syndrome and septic shock[1]. During the acute phase of infection, myeloid cells, including monocytes and macrophages, are the predominant immune cell type in the lungs of COVID-19 patients and play a major role in the pathogenicity of the disease[2,3]. However, contrasting observations have been reported regarding the involvement of myeloid cells in the development of cytokine storms vs. immunosuppression[4,5] and the overactive or deficient type I IFN response generated by myeloid cells in the lungs and in peripheral blood[6–11]. Despite these inconsistent reports, most studies have observed dysregulated innate immune responses and reduced expression of human leukocyte antigen DR (HLA-DR) by circulating myeloid cells, which is considered a marker of immune suppression[10,12–16].

Monocytes are blood-circulating, phagocytic, innate immune leukocytes with important functions in pathogen sensing, and innate and adaptive immune response activation during viral infection[17]. Despite their heterogeneity[18], human monocytes are broadly classified into three subsets based on the expression of CD14 and CD16, i.e., classical (CD14+CD16-), intermediate (CD14+CD16+), and nonclassical

[1]Department of Infectious Diseases, Faculty of Medicine, Imperial College London, London, UK. [2]Wellcome Sanger Institute, Wellcome Genome Campus, Hinxton, Cambridge, UK. [3]Department of Immunology and Inflammation, Faculty of Medicine, Imperial College London, London, UK. [4]Aix Marseille Université, CNRS, INSERM, Centre d'Immunologie de Marseille-Luminy, Marseille, France. ✉e-mail: m.dominguez-villar@imperial.ac.uk

(CD14[low]CD16[+]) monocytes[17]. During viral infection, circulating monocytes infiltrate affected tissues and differentiate into inflammatory macrophages and dendritic cells (DCs)[19], contributing to pathogen clearance and tissue regeneration.

Here, we examine the phenotype and functionality of the main monocyte population in humans, i.e., classical CD14[+] monocytes, in patients with COVID-19 during the acute phase of disease and compare them to those of healthy individuals. We find that ex vivo isolated CD14[+] monocytes from mild and moderate COVID-19 patients are phenotypically different from monocytes from healthy individuals, displaying differential expression of costimulatory and inhibitory receptors, MHC molecules and a dysfunctional metabolic profile that is accompanied by decreased ex vivo NF-κB activation, while maintaining an intact type I IFN antiviral response. Subsequent pathogen sensing ex vivo led to a state of functional unresponsiveness of COVID-19 monocytes that was associated transcriptionally with that of an endotoxin-induced tolerance signature, characterized by the decrease in canonical innate immune functions, including the expression of activation markers and pro-inflammatory cytokine production. In addition, their gene expression signature and function switched from canonical innate immune functions to a pro-thrombotic phenotype characterized by increased

expression of pathways involved in immunothrombosis and increased capacity to form cell aggregates with platelets. These results provide a potential mechanism by which innate immune dysfunction in COVID-19 contributes to disease progression.

## Results

### Phenotypic alterations in COVID-19 monocytes

Global alterations in innate immune cell phenotypes have been identified in severe COVID-19[11,20–22]. As the main human monocyte population, we focused on deeply characterizing the ex vivo phenotype of classical CD14[+] monocytes in uninfected healthy individuals and patients with COVID-19 presenting with mild or moderate symptoms (1-2 or 3-4 WHO ordinal scale for COVID-19 severity, respectively) during the acute phase of disease (Dataset 1). The battery of markers examined by high dimensional flow cytometry included MHC molecules and costimulatory and coinhibitory receptors (Fig. 1). Dimensionality reduction tools demonstrated that while some overlap in the global phenotypes was observed among the three study groups, monocytes from healthy individuals were clearly distinct from both mild and moderate COVID-19 on a tSNE plot (Fig. 1a, n = 15 individuals per group). In addition, COVID-19 monocytes could be distinguished

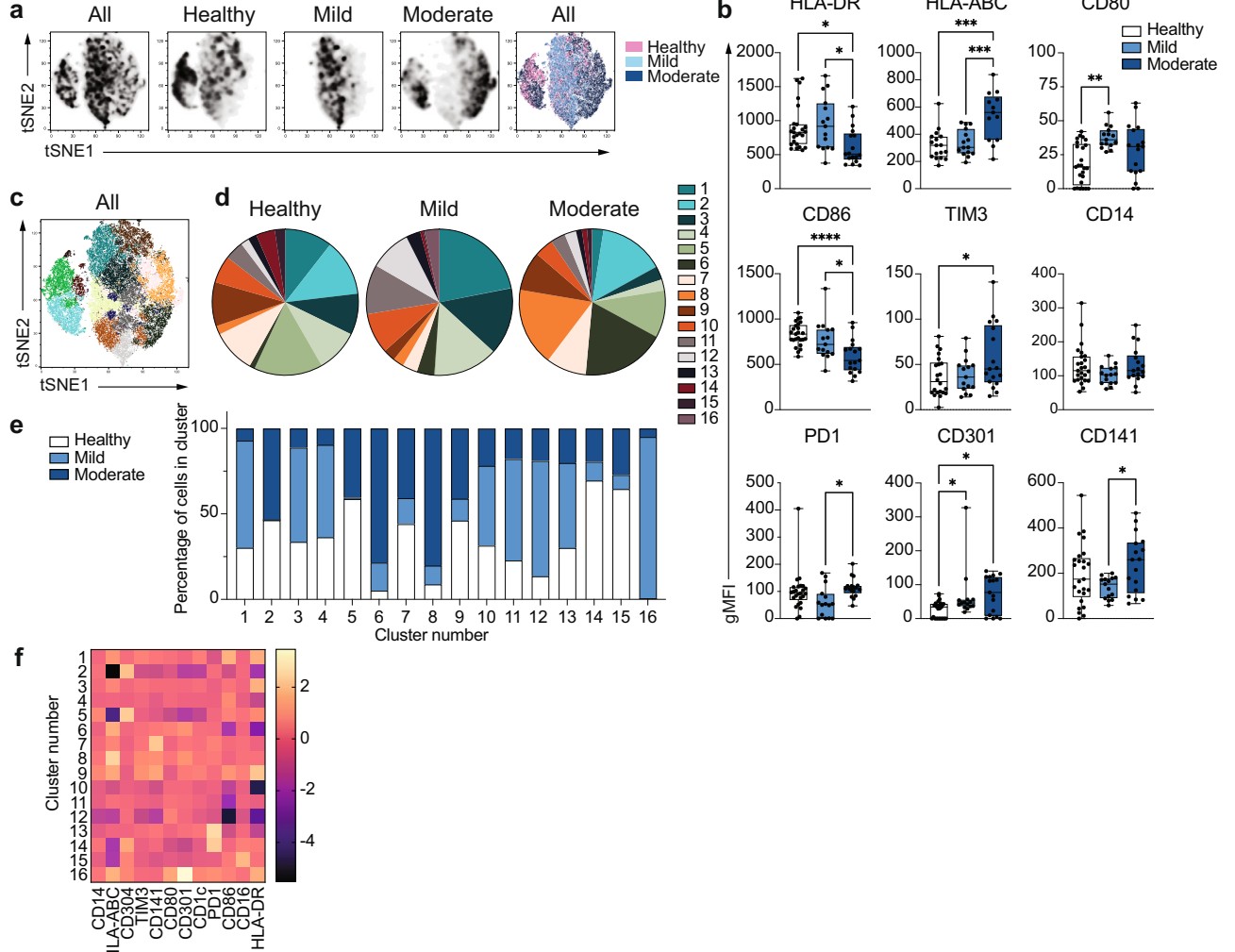

**Fig. 1 | Unique phenotype of COVID-19 monocytes. a** tSNE plots obtained from a concatenated sample consisting of CD14[+] classical monocytes from *n* = 15 healthy individuals, *n* = 15 mild and *n* = 15 moderate COVID-19 patients. **b** Box-and-whisker plots summarizing the median gMFI of the receptors analyzed (*n* = 25 healthy, *n* = 15 mild and *n* = 17 moderate COVID-19 individuals). **c** tSNE plots depicting the cell clusters identified by Phenograph from the concatenated sample in **a**. **d** Pie charts show the fraction of cells within each identified cell cluster in each patient group. **e** Bar graphs show the distribution (percentage) of cells from each patient group in each identified cell cluster. **f** Heatmap of the expression of receptors per cell cluster displayed as modified z-scores using median values. One-way ANOVA with Tukey's correction for multiple comparisons for **b**. *$p < 0.05$, **$p < 0.005$, ***$p < 0.001$, ****$p < 0.0001$. Source data are provided as a Source Data file.

based on disease severity, with main cell clusters for both disease severity groups mapping separately on the tSNE plots. Moderate COVID-19 monocytes expressed decreased levels of HLA-DR, in agreement with previous reports[10,16], but in contrast, they displayed increased expression of HLA-ABC compared to both mild disease and uninfected individuals, suggesting a skewed trend towards class I antigen presentation (Fig. 1b, n = 25 healthy, n = 15 mild and n = 17 moderate COVID-19 patients). In addition, moderate COVID-19 monocytes expressed increased levels of the c-type lectin CD301. The decreased expression of the costimulatory receptor CD86 on moderate COVID-19 monocytes compared to healthy and mild COVID-19 patients, the increased expression of the inhibitory receptors TIM-3[23] compared to healthy individuals, and PD1[24] compared to mild COVID-19 monocytes suggest an altered activation profile skewed towards an inhibitory phenotype. In addition, significant differences in the expression of certain markers were found between mild and moderate COVID-19 monocytes. For example, downregulation of HLA-DR and CD86 and upregulation of TIM-3 and HLA-ABC compared to healthy monocytes were only significant in moderate but not in mild COVID-19 monocytes, and the increased expression of CD80 in mild COVID-19 compared to healthy monocytes was not apparent in moderate COVID-19 (Fig. 1b). These results suggest a more profound dysfunction in moderate than in mild COVID-19 monocytes, and they were further confirmed in a second cohort of healthy individuals, mild and moderate COVID-19 patients (Supplementary Fig. 1). Moreover, in agreement with the altered HLA and costimulatory receptor profile of COVID-19 monocytes, we observed an impairment in their capacity to activate SARS-CoV-2-specific CD4+ and CD8+ T cells (Supplementary Fig. 2). Thus, CD14+ monocytes from both mild and moderate COVID-19 patients were able to efficiently activate SARS-CoV-2-specific CD8+ T cells upon UV-inactivated SARS-CoV-2 stimulation. However, only CD14+ monocytes from healthy individuals were able to trigger the activation of SARS-CoV-2-specific CD4+ T cells.

To further define and quantify the phenotypic differences observed between healthy individuals and COVID-19 patients, we applied clustering algorithms using the 12 phenotypic markers previously examined. Cell clustering identified 16 different subpopulations of monocytes that were distinctively distributed in healthy and COVID-19 monocytes (Fig. 1c, d, Dataset 2), with 11 clusters containing more than 88% of the total cells analyzed (Supplementary Fig. 3). Interestingly, expansions of specific monocyte subpopulations were different in mild and moderate COVID-19 monocytes. Thus, statistically significant differences in the size of clusters 1, 3, 4, 6, 8, 11, 12, and 15 were found among clinical groups (Supplementary Fig. 4). In particular, both mild and moderate COVID-19 patients had reduced frequency of clusters 3 and 15 compared to that of healthy monocytes, while moderate patients had a significant increased size of clusters 6 and 8 compared to both mild patients and healthy individuals. Finally, differences in the size of specific clusters were observed between mild and moderate COVID-19 patients. Thus, mild patients had significantly higher frequency of cells in clusters 1, 4, 11, and 12 as compared to both moderate patients and healthy individuals (except for cluster 4, which was only significantly elevated as compared to moderate COVID-19 patients, Fig. 1d and Supplementary Fig. 4). As a consequence, the distribution of cells from healthy, mild and moderate COVID-19 monocytes was clearly different in each cluster, and while some cell clusters were composed of cells from all disease groups, such as clusters 10, 11 and 13, other clusters predominantly contained cells from one or two particular disease groups. For example, clusters 1, 3, 4, 12, and 16 were predominantly composed of cells from mild patients, while clusters 6 and 8 predominantly contained moderate COVID-19 monocytes and were almost absent in monocytes from healthy individuals (Fig. 1e). Normalized expression levels of the markers defining each cluster demonstrated that the phenotype of cluster 6 was mostly driven by downregulation of CD86 and HLA-DR, while that of cluster 8

was mostly driven by the increased expression of HLA-ABC (Fig. 1f). Collectively, these results reveal that distinct populations of circulating monocytes are enriched in mild and moderate COVID-19 patients.

## Metabolic dysfunction in COVID-19 monocytes

The fundamental differences in the phenotype of moderate COVID-19 monocytes compared to that of healthy individuals led us to investigate in depth the gene expression profile of ex vivo isolated classical CD14+ monocytes from patients with moderate COVID-19 and compare them with those of healthy individuals (Fig. 2). Principal component analysis (PCA) applied to examine the global distribution of gene expression profiles from COVID-19 monocytes (n = 10) and healthy individuals (n = 6) demonstrated a clear separation between groups along PC1 (Fig. 2a), with genes encoding a number of soluble factors, chemokines and HLA class II molecules as the main genes contributing to the separation between healthy and COVID-19 monocytes (Supplementary Fig. 5). Differential gene expression analysis yielded 422 upregulated and 187 downregulated genes (≥1.5-fold change, FDR < 0.05) in COVID-19 monocytes compared to healthy controls (Fig. 2b). We used these genes to perform a pathway enrichment analysis with XGR[25] and pathway annotations from Reactome to gain insight on potential pathways differentially expressed in COVID-19 monocytes (Supplementary Fig. 6 and Dataset 3). Interestingly, pathway enrichment identified glycolysis as the most enriched pathway in COVID-19 monocytes, and also included metabolism of lipids and lipoproteins among the statistically significant pathways (Supplementary Fig. 6). Other significantly enriched pathways included interferon signaling and cytokine signaling, results that are in agreement with previous reports on the role of these two pathways in COVID-19 pathogenesis[6,16,22] (Supplementary Fig. 6 and Dataset 3).

We subsequently examined the directionality of expression of the enriched pathways by analyzing downregulated genes and upregulated genes separately (Fig. 2c). Pathway enrichment analysis of genes significantly upregulated (≥1.5-fold change, FDR < 0.05) in COVID-19 compared to healthy individuals demonstrated a significant increase in the metabolism of a number of lipids, including sphingolipids, phospholipids, and lipoproteins. Other upregulated pathways in COVID-19 monocytes included interferon signaling, cytokine signaling, and transmembrane transport of small molecules (Fig. 2c). Heatmap showing the top 40 upregulated genes from the enriched pathways demonstrated a somewhat variable expression patterns among COVID-19 monocytes and included a number of type I interferon-stimulated genes (IFI27, IFITM2, IFI6, IFITM3, MX1), metabolic enzymes (ASAH1, CYP27A1, SGPP2, SPHK1) and others (Fig. 2d). Of note, the highest expressed IFN-related gene was IFI27, which has been suggested as a biomarker of early SARS-CoV-2 infection[26]. The increased type I IFN gene signature in COVID-19 monocytes was confirmed by the increased ex vivo phospho-IRF3 protein expression in moderate COVID-19 patients compared to healthy individuals (Fig. 2e, f) and by the increased expression of IFITM2 as an IFN-stimulated gene, measured by real-time PCR in an expanded cohort of mild and moderate COVID-19 patients that included samples from those individuals whose monocytes were subjected to RNA-seq (n = 6 healthy individuals and n = 10 moderate COVID-19 patients) plus additional samples of healthy individuals (n = 1), mild (n = 7) and moderate patients (n = 3, Fig. 2g). NFκB activation was examined ex vivo indirectly by IκBα expression (Fig. 2h, i) and directly by phosphorylation of the p65 NFκB subunit (Fig. 2j, k), as a readout for cytokine signaling[27,28]. While monocytes from mild COVID-19 patients displayed a decrease in the expression of IκBα compared to that of healthy individuals (Fig. 2i), monocytes from moderate COVID-19 patients did not. Furthermore, neither mild nor moderate COVID-19 monocytes displayed an increased expression of phospho-p65 NFκB (Fig. 2k). This observation suggests that other additional mechanisms may be regulating the activation of NFκB, and

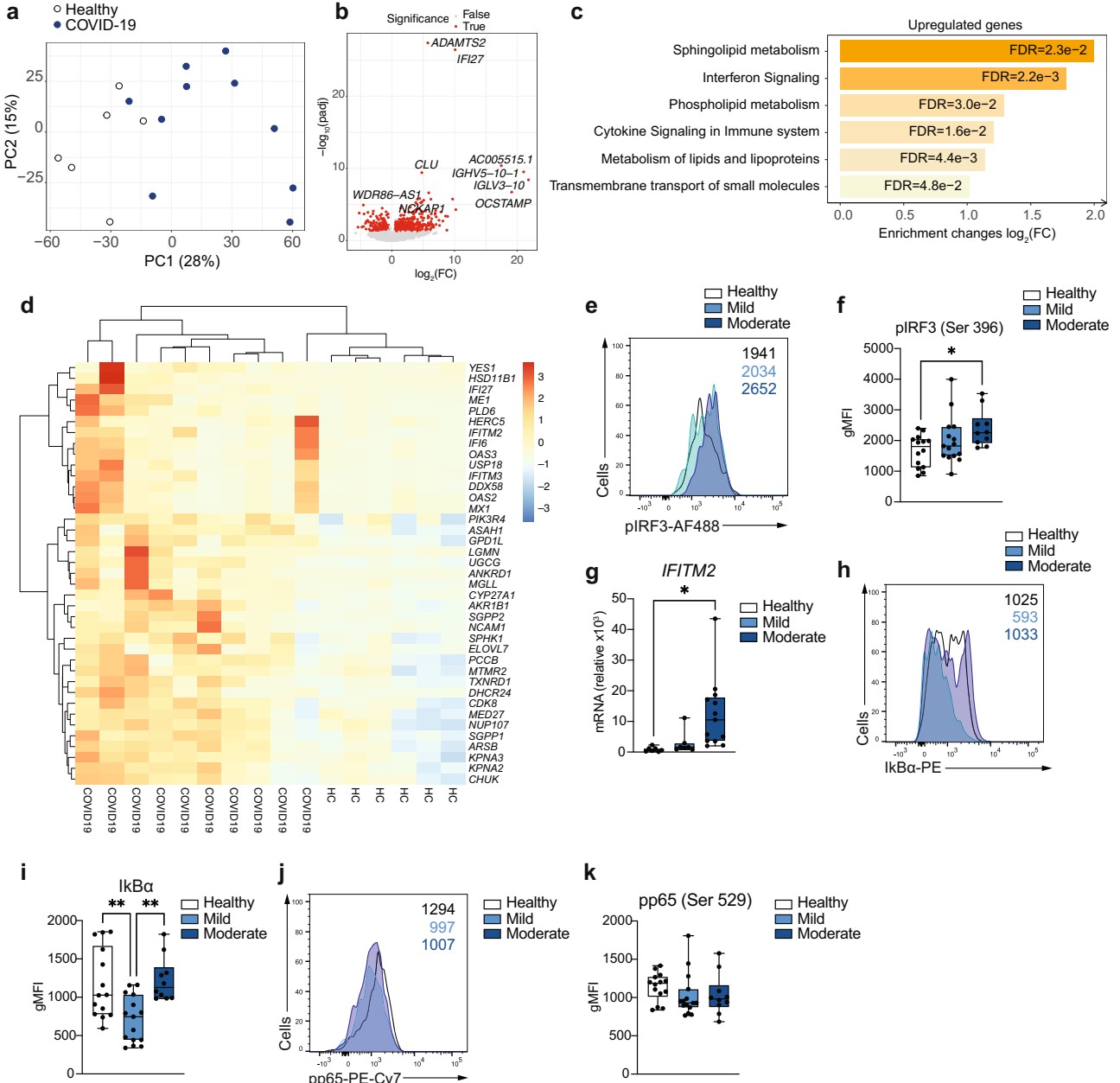

**Fig. 2 | Gene expression signature of COVID-19 monocytes ex vivo. a** Principal component analysis of the gene expression data computed from all genes from ex vivo healthy individual (white dots) and moderate COVID-19 (blue dots) monocyte samples. The variance explained by each component is stated in brackets. **b** Volcano plot of differentially expressed genes for ex vivo COVID-19 vs. healthy monocytes. Genes with fold change ≥1.5 and FDR < 0.05 are shown in red. **c** Bar plots depict significantly enriched (FDR < 0.05) pathways from Reactome for COVID-19 vs. healthy individual monocytes using upregulated genes (≥1.5 fold increase, FDR < 0.05). Fold enrichment plotted as $\log_2$ (FC) and bars labelled with the adjusted *p*-value. **d** Heatmap of significantly upregulated genes in COVID-19 vs. healthy monocyte that are members of the pathways in **c**. Gene expression values are scaled by row, with relatively high expression indicated in red and relatively low expression in blue. Both rows and columns are clustered using Euclidean distance and Ward's method. Representative example (**e**) and summary gMFI (**f**) of

phosphorylated (p)-IRF3 (Ser 396) expression for healthy ($n = 14$), mild ($n = 15$) and moderate ($n = 10$) COVID-19 monocytes. **g** *IFITM2* relative gene expression (to *GAPDH*) measured by real-time PCR in sorted CD14$^+$ monocytes from healthy individuals ($n = 7$), mild ($n = 7$) and moderate ($n = 13$) COVID-19 patients. Representative example (**h**) and summary gMFI (**i**) of IκBα expression in healthy individuals ($n = 14$), mild ($n = 15$) and moderate ($n = 10$) COVID-19 monocytes. Representative example (**j**) and summary gMFI (**k**) of pNFκB p65 expression in healthy individuals ($n = 14$), mild ($n = 15$) and moderate ($n = 10$) COVID-19 monocytes. In **e**, **h**, and **k**, numbers in histograms represent the gMFI of healthy (black), mild (light blue), and moderate (dark blue) COVID-19 patients. In **f**, **g**, **i** and **k**, boxes extend from the 25th to the 75th percentiles and whiskers extend up to the maximum and down to the minimum values. The horizontal line within the boxes represents the median. One-way ANOVA with Tukey's test for multiple comparisons in **f**, **g**, **i**, and **k**. *$p < 0.05$, **$p < 0.005$. Source data are provided as a Source Data file.

that NFκB-driven cytokine responses may be altered in patients with COVID-19, in agreement with the lack of increased pro-inflammatory cytokine expression by COVID-19 monocytes (Fig. 2c, Dataset 4) and with previous single cell transcriptomic data of acute COVID-19 PBMC[29]. Moreover, several of the genes contributing to the

"Cytokine signaling" pathway enrichment (Fig. 2c) were interferon-stimulated genes (Dataset 4).

We subsequently selected the set of significantly downregulated genes in COVID-19 monocytes (≥1.5 fold decrease, FDR < 0.05) to perform pathway enrichment (Fig. 3). The only pathway that was

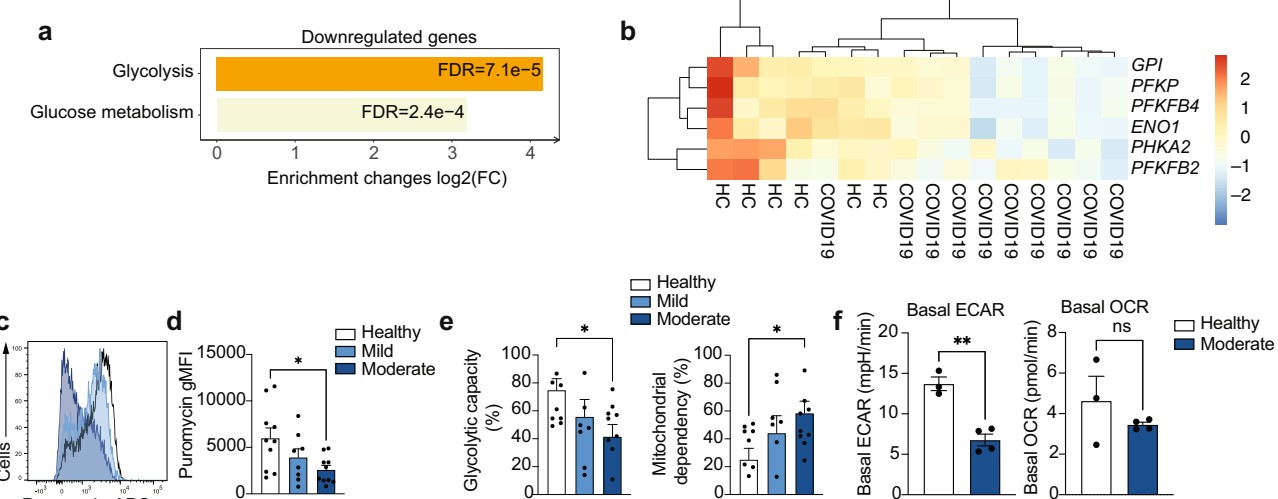

**Fig. 3 | Altered expression of glycolysis-related genes in COVID-19 monocytes ex vivo. a** Bar plots depict significantly enriched (FDR < 0.05) pathways from Reactome for COVID-19 vs. healthy individual monocytes, using downregulated genes in COVID-19 vs. healthy (≥1.5 fold decrease, FDR < 0.05). The fold enrichment is plotted on the x-axis as $\log_2$(FC) and the bars labelled with the adjusted p-value. **b** Heatmap of the significantly downregulated genes in COVID-19 vs. healthy monocytes that are members of the pathways in **i**. Gene expression values are scaled by row, with relatively high expression indicated in red and relatively low expression in blue. Both rows and columns are clustered using Euclidean distance and Ward's method. Representative example of ex vivo expression of puromycin in CD14[+] monocytes measured by flow cytometry (**c**) and summary (**d**) of puromycin gMFI on healthy individuals ($n = 10$), mild ($n = 8$) and moderate ($n = 10$) COVID-19 monocytes (right). **e** Glycolytic capacity and mitochondrial dependency of monocytes from healthy individuals ($n = 10$), mild ($n = 8$) and moderate ($n = 10$) COVID-19 ex vivo. **f** Basal extracellular acidification rate (ECAR) and basal oxygen consumption rate (OCR) measured in sorted CD14[+] monocytes from healthy individuals ($n = 3$) and COVID-19 patients ($n = 4$). The data in **d**, **e**, and **f** are shown as mean ± s.e.m. One-way ANOVA with Tukey's test for multiple comparisons in **d** and **e**, and unpaired, two-tailed t-test in **f**. *$p < 0.05$, **$p < 0.005$. Source data are provided as a Source Data file.

significantly downregulated in COVID-19 monocytes was glycolysis (Fig. 3a and Dataset 5), with decreased expression of a number of enzymes involved in glucose degradation, including *PFKP*, *ENO1*, *PFKB4* and others (Fig. 3b). This metabolic profile with increased metabolism of lipids (Fig. 2c) and decreased glycolysis was unexpected, as glycolysis is an important driver of innate immune cell function during the recognition of pathogens[30]. To confirm these gene expression data, we used SCENITH[TM31] to metabolically profile CD14[+] monocytes from COVID-19 patients and healthy controls ex vivo. SCENITH[TM] uses protein synthesis as a measurement of global metabolic activity. Puromycin incorporation is used as a reliable readout of protein synthesis levels (and therefore metabolic activity) in vitro and in vivo. In agreement with the pathway enrichment results, ex vivo puromycin incorporation was significantly decreased in moderate COVID-19 monocytes (Fig. 3c, d) compared to healthy individuals, suggesting decreased metabolic activity. Moreover, the glycolytic capacity of COVID-19 monocytes was significantly decreased in moderate patients and correlated with disease severity (Fig. 3e), and this was accompanied by a concomitant increase in mitochondrial dependency in monocytes from moderate COVID-19 patients (Fig. 3e). The decreased metabolic activity and glycolytic capacity were further confirmed by Seahorse analysis of extracellular acidification rate (ECAR) and oxygen consumption rate (OCR) as readouts for glycolysis and oxidative phosphorylation, respectively (Fig. 3f).

These data suggest that monocytes from COVID-19 patients with moderate disease display a distinct gene expression signature, characterized by an impaired metabolic profile that is accompanied by decreased NFκB activation, and maintenance of an intact type I IFN antiviral response.

**COVID-19 monocytes display impaired pathogen sensing ex vivo**
The dysfunctional metabolic profile with a downregulation of glycolysis and the defective activation of NFκB, both pathways heavily involved in the activation of canonical innate immune cell functions upon virus encounter[28,30], led us to examine the functional capacity of

monocytes to sense and respond to SARS-CoV-2 ex vivo (Fig. 4). Upon UV-inactivated SARS-CoV-2 stimulation ($10^6$ viral particles per $10^6$ cells for 20 h), CD14[+] monocytes from healthy individuals displayed a significant increase in both TNF and IL-10 production (Fig. 4a, b). However, COVID-19 monocytes significantly produced less TNF as compared to healthy monocytes, while no differences were observed in IL-10 expression (Fig. 4a, b). The defect in TNF production upon stimulation was not SARS-CoV-2-specific, as stimulation with inactivated common cold coronaviruses ($10^6$ viral particles per $10^6$ cells for 20 h) or bacterial lipopolysaccharide (LPS, 100 ng/ml for 20 h) also led to significantly reduced TNF production compared to monocytes from healthy individuals (Fig. 4c). In addition, the expression of CD40 (Fig. 4d, e), which is important for monocyte effector function and is upregulated after virus sensing[32], was increased in monocytes from healthy individuals but not in COVID-19 monocytes. This decreased expression was confirmed after stimulation with common cold coronaviruses or LPS (Fig. 4f), suggesting that the activation defects in COVID-19 monocytes in response to pathogen sensing were not specific to SARS-CoV-2. In addition to CD40, we also examined the expression of other cell surface receptors involved in antigen presentation and activation of T cells (Fig. 4g). HLA-DR expression levels were not further upregulated upon SARS-CoV-2 stimulation in any of the patient groups, and stimulation still maintained the differences in expression observed ex vivo among groups (Fig. 4h). Moreover, while CD80 was significantly upregulated in healthy, mild and moderate COVID-19 monocytes after SARS-CoV-2 stimulation (Fig. 4g, i), only healthy monocytes increased the expression of CD86 after stimulation (Fig. 4j). These results were confirmed in additional in vitro stimulation with isolated CD14[+] monocytes (Supplementary Fig. 7).

The apparent unresponsiveness of COVID-19 monocytes to pathogen sensing was accompanied by altered metabolic reprogramming. Innate immune cells that sense pathogens increase the rate of glycolysis over mitochondrial oxidative phosphorylation to enable fast energy availability[33–35]. However, COVID-19 monocyte energetic profile measured by SCENITH[TM] did not increase upon LPS stimulation, unlike

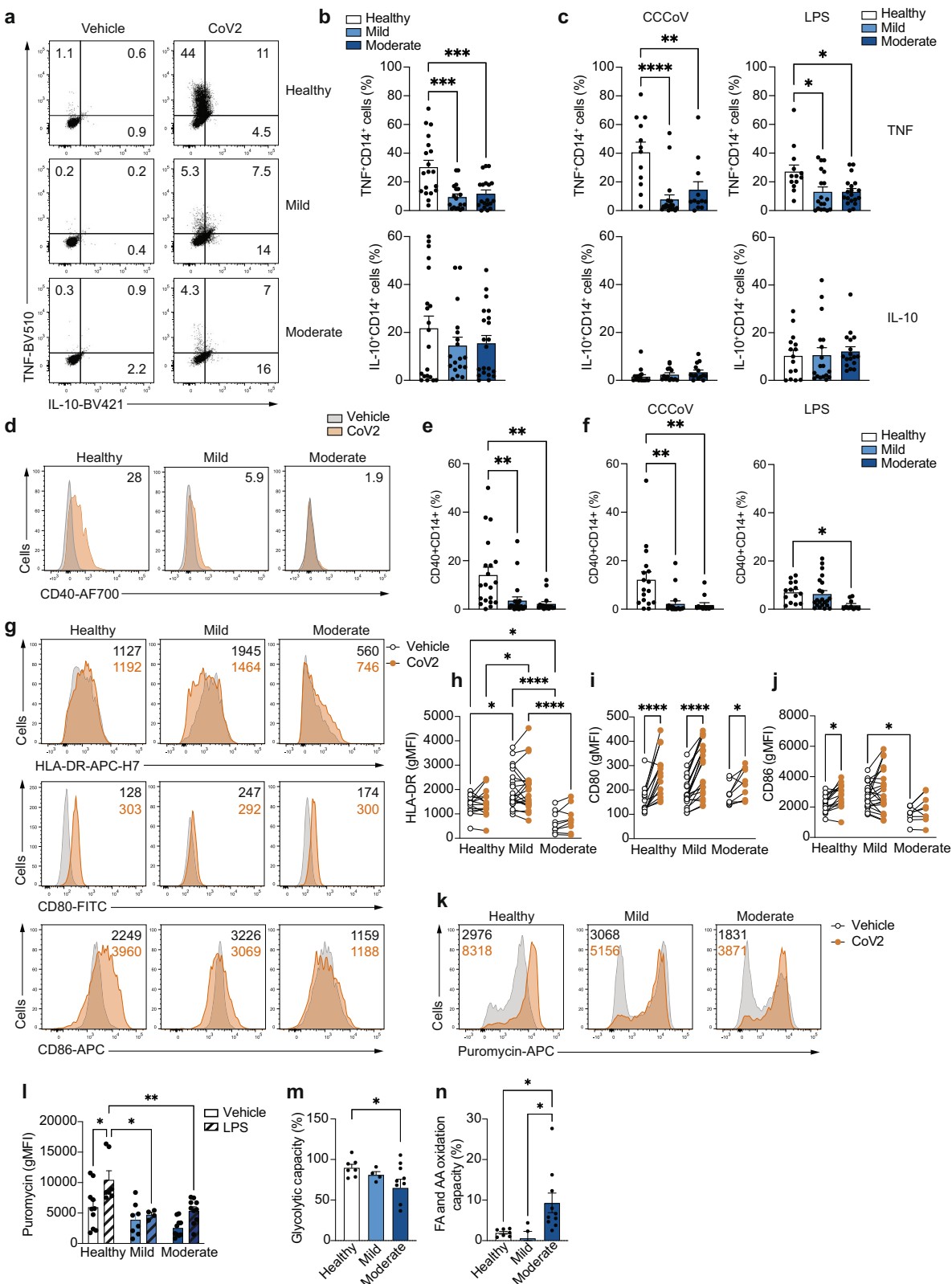

that of healthy monocytes (Fig. 4k, l). Moreover, when stimulated with LPS, moderate COVID-19 monocytes showed a decreased glycolytic capacity (Fig. 4m) and an increase in fatty acid and amino acid oxidation capacity (Fig. 4n) compared to healthy monocytes, that correlated with a slight but significant decrease in glucose dependency and an increase in mitochondrial dependency compared to monocytes from healthy individuals (Supplementary Fig. 8). These data are in

agreement with the enriched metabolic pathways from RNA-seq data (Fig. 2c, h). Seahorse experiments confirmed the defect in glycolysis in stimulated monocytes from COVID-19 patients (Supplementary Fig. 9). In summary, monocytes from COVID-19 patients display a profound defect in pathogen sensing ex vivo that is more evident in moderate than in mild patients and is characterized by an impairment in pro-inflammatory cytokine production, expression of activation-related

**Fig. 4 | Impaired ex vivo pathogen sensing by COVID-19 monocytes.** Representative example (**a**) and summary (**b**) of TNF and IL-10 production by monocytes from healthy individuals (*n* = 19), mild (*n* = 18), and moderate COVID-19 patients (*n* = 19) after ex vivo stimulation with SARS-CoV-2. **c** Summary of percentage of TNF- and IL-10-producing monocytes after stimulation with a mixture of heat-inactivated common cold coronaviruses (CCCoV) or LPS in healthy individuals (*n* = 12 for CCCoV and *n* = 13 for LPS), mild (*n* = 21 for CCCoV and *n* = 18 for LPS) and moderate (*n* = 12 for CCCoV and *n* = 19 for LPS) COVID-19 patients. Representative histograms (**d**) and summary (**e**) of CD40 expression in healthy individual (*n* = 20), mild (*n* = 22), and moderate (*n* = 16) COVID-19 monocytes stimulated with vehicle (grey) or SARS-CoV-2 (orange). Numbers represent percentage of CD40⁺ monocytes relative to vehicle-stimulated cells. **f** Summary of percentage of CD40⁺CD14⁺ cells after stimulation with CCCoV or LPS in healthy individuals (*n* = 17 for CCCoV and *n* = 14 for LPS), mild (*n* = 18 for CCCoV and *n* = 22 for LPS) and moderate (*n* = 13 for CCCoV and *n* = 10 for LPS) COVID-19 patients. Representative histograms (**g**)

and summary gMFI of HLA-DR (**h**), CD80 (**i**) and CD86 (**j**) expression of CD14⁺ monocytes from healthy individuals (*n* = 15), mild (*n* = 22) and moderate (*n* = 9) COVID-19 patients stimulated with vehicle (white) or SARS-CoV-2 (CoV2, orange). Lines link paired samples. Representative histogram (**k**) and summary (**l**) of monocyte energetic status measured by puromycin expression (gMFI) of monocytes from healthy individuals (*n* = 10), mild (*n* = 8) or moderate (*n* = 10) COVID-19 patients stimulated with vehicle (open bars) or LPS (striped bars). Glycolytic capacity (**m**) and fatty acid and amino acid oxidation capacity (**n**) of CD14⁺ monocytes from healthy individuals (*n* = 7), mild (*n* = 4), and moderate (*n* = 9) COVID-19 patients stimulated with LPS. The data in **b**, **c**, **e**, **f**, **l**, **m** and **n** are shown as mean ± s.e.m. One-way ANOVA with Tukey's correction for multiple comparisons in **b**, **c**, **e**, **f**, **m** and **n**. Two-way ANOVA with Tukey's correction for multiple comparisons in **h**, **i**, **j** and **l**. *$p < 0.05$, **$p < 0.005$, ***$p < 0.001$, ****$p < 0.0001$. Source data are provided as a Source Data file.

receptors and metabolic rewiring upon secondary SARS-CoV-2 stimulation.

## Pro-thrombotic gene expression signature of COVID-19 monocytes

To globally characterize the gene expression signature of activated monocytes in COVID-19, we performed RNA-seq of isolated monocytes from healthy individuals and patients with moderate COVID-19 stimulated for 20 h with UV-inactivated SARS-CoV-2 as in Fig. 4 (Fig. 5). PCA clearly separated COVID-19 from healthy monocytes, although some healthy monocytes clustered with COVID-19 in the principal component space (Fig. 5a, Supplementary Fig. 10). Quantification of differentially expressed genes yielded 1,437 upregulated and 2,073 downregulated genes in activated COVID-19 compared to activated healthy monocytes (≥1.5 fold change, FDR < 0.05, Fig. 5b). Pathway enrichment of differentially expressed genes (≥1.5 fold change vs. healthy monocytes, FDR < 0.05) using XGR software and the Reactome pathway database demonstrated a number of expected pathways involved in the innate immune response to pathogens, including type I IFN signaling, cytokine signaling, interactions between lymphoid and non-lymphoid cells, NLR sensing, etc. (Supplementary Fig. 11 and Dataset 6). However, when we focused our analysis on pathways enriched in upregulated genes in activated COVID-19 monocytes compared to activated healthy monocytes, the most significantly enriched pathways were involved in or closely related to hemostasis and coagulation, including integrin signaling, extracellular matrix organization, signaling by PDGF, interactions with activated platelets and general hemostasis (Fig. 5c and Dataset 7). Integrin receptors are used by cells to interact with other cells and with the extracellular matrix, by binding numerous matrix proteins including collagen, actin and laminin, being also involved in hemostasis and platelet aggregation[36]. In addition, monocytes actively bind to platelets forming pro-thrombotic aggregates in inflammatory and vascular pathologies[37,38]. Monocytes from COVID-19 patients expressed increased levels of various collagen subunits (*COL1A1*, *PLOD2*, *COL6A3*, *COL6A1*), enzymes involved in collagen triple helix synthesis (*COL-GALT1*) and a number of matrix metalloproteinases (*MMP1*, *MMP2*, *MMP14*, Fig. 5d), which are not only involved in extracellular matrix remodeling, but they have also been implicated in contributing directly to platelet activation and priming for aggregation[39,40]. These results are in agreement with the clinical observations of hypercoagulability and acquired coagulopathies in patients with COVID-19[41–44], and suggest that monocytes from moderate COVID-19 patients upregulate a pro-thrombotic gene expression signature upon secondary SARS-CoV-2 sensing.

Interestingly, downregulated pathways in stimulated COVID-19 monocytes compared to stimulated healthy donor monocytes included most of the canonical immunological functions expected for innate immune cells upon virus sensing, i.e., interferon signaling, RIG-I/

MDA5-mediated induction of interferons, activation of TCR signaling in T cells, innate immune functions and interactions with non-lymphoid cells (Fig. 5e and Dataset 8). The majority of the top 40 genes significantly downregulated in COVID-19 monocytes from these downregulated pathways consisted of different interferons (*IFNA1*, *IFNA2*, *IFNA14*, and *IFNB1*), interferon-stimulated genes (*IFIT3*, *ISG15*, *IFIT2*, *ISG20*, *IRF7*, and *MX2*) and pathogen-sensing receptors (*TLR7*, *AIM2*, Fig. 5f). This gene signature was functionally confirmed by examining the activation pattern of IRF3 in response to LPS in monocytes from healthy individuals and patients with mild and moderate COVID-19 (Fig. 5g). While healthy and mild COVID-19 monocytes significantly increased the expression of the phosphorylated form of IRF3 upon LPS stimulation compared to baseline levels, monocytes from moderate patients did not. This inability to activate IRF3 correlated with decreased expression of the interferon-stimulated gene *IFITM2*, examined in an expanded cohort of healthy, mild and moderate COVID-19 monocytes after stimulation with SARS-CoV-2 (Fig. 5h). Of note, examination of NFκB p65 activation, as a main transcription factor involved in cytokine signaling in innate cells, demonstrated a defective activation in both mild and moderate COVID-19 as compared to healthy individuals (Fig. 5i).

These findings in COVID-19 monocytes are consistent with an unexpected switch from canonical innate immune functions to a pro-thrombotic phenotype and potential cross-talk with other cells involved in hemostasis, which suggests that activated monocytes may contribute to COVID-19 severity by actively impacting hemostasis and by a reduction in innate immune functions necessary for efficient virus clearance.

## Monocytes from COVID-19 patients are functionally pro-thrombotic

In order to functionally confirm the pro-thrombotic gene expression signature of monocytes from COVID-19 we performed in vitro assays to test the capacity of monocytes to form monocyte-platelet aggregates (MPA, Fig. 6), which are an important initiation factor in the generation of thrombi[45–47]. Isolated monocytes from healthy individuals and patients with mild or moderate COVID-19, either unstimulated or stimulated with UV-inactivated SARS-CoV-2 ex vivo, were co-cultured with freshly isolated platelets from a healthy individual to rule out the possibility that differential activation of platelets from healthy individuals and COVID-19 patients would lead to confounding results on the pro-thrombotic capacity of the monocytes. After 20 h of monocyte-platelet co-culture, we measured the generation of MPA by determining the expression of CD41 in monocytes, which is a standard method to identify MPA (CD41 is a marker of megakaryocytes and it is not expressed on monocytes[48]). Monocytes from patients with moderate COVID-19 aggregated significantly more platelets that those of healthy individuals or patients with mild disease, functionally supporting the pro-thrombotic signature obtained in the RNA-seq analysis

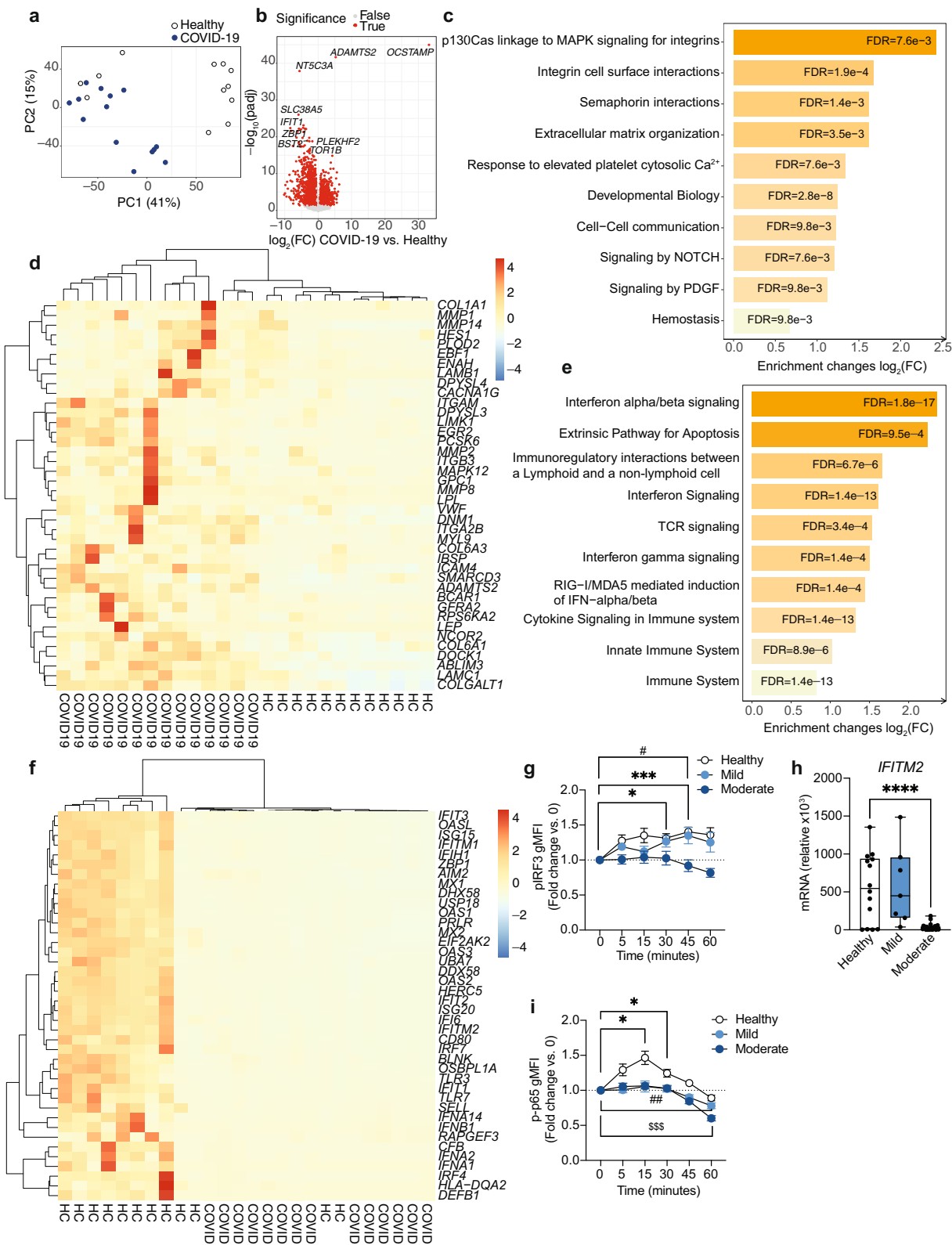

(Fig. 6a–c). Interestingly, the increase in MPA formation was observed after ex vivo stimulation with UV-inactivated SARS-CoV-2 (Fig. 6a, c) but also in ex vivo unstimulated monocytes, which was initially surprising due to the lack of hemostasis-related pathways being significantly enriched in the ex vivo RNA-seq datasets (Figs. 2 and 3). However, upon examination of the RNA-seq analysis from ex vivo unstimulated monocytes, we identified a number of significantly upregulated genes in COVID-19 monocytes compared to healthy individuals that belonged to pathways that included "Hemostasis" and "Platelet activation, signaling and aggregation" (Dataset 4). Enrichment of none of these two pathways was statistically significant when considering the adjusted *p*-value, but upregulation of the individual genes was, which could be due to the small number of RNA-seq samples and the large number of pathways to adjust the *p*-value for.

**Fig. 5 | Gene expression signature of SARS-CoV-2-stimulated COVID-19 monocytes. a** Principal component analysis of all genes from healthy (white) and moderate COVID-19 (blue) monocytes stimulated with SARS-CoV-2. The variance explained by each component is stated in brackets. **b** Volcano plots of differentially expressed genes for activated COVID-19 vs. activated healthy monocytes. Red shows genes with fold change ≥1.5 and FDR < 0.05. **c** Bar plots depict the top 10 significantly enriched (FDR < 0.05) pathways for stimulated COVID-19 vs. healthy individual monocytes using upregulated genes (≥1.5 fold increase, FDR < 0.05). Fold enrichment is plotted as $\log_2$(FC) and bars labelled with the adjusted $p$-value. **d** Heatmap of the top 40 significantly upregulated gene members of the pathways in **c. e** Bar plots depict the top 10 significantly enriched (FDR < 0.05) pathways for stimulated COVID-19 vs. healthy individual monocytes, using downregulated genes (≥1.5 fold decrease, FDR < 0.05), plotted as $\log_2$(FC) and bars labelled with the adjusted $p$-value. **f** Heatmap of the top 40 significantly downregulated genes in stimulated COVID-19 vs. healthy individual monocytes that are members of the pathways in **e. g** Phospho-IRF3 (Ser 396) expression (fold change to baseline gMFI)

for healthy ($n = 14$), mild ($n = 15$) and moderate ($n = 10$) COVID-19 monocytes stimulated with LPS (mean ± s.e.m.). **h** *IFITM2* gene expression (relative to *GAPDH*) measured by real-time PCR and stimulated monocytes from healthy individuals ($n = 14$), mild ($n = 7$), and moderate ($n = 23$) COVID-19 patients. Boxes extend from the 25th to the 75th percentiles, the horizontal line within the boxes shows the median, and the whiskers extend from the minimum to the maximum values. **i** Phospho-NFκB p65 (Ser 529) expression (fold change to baseline gMFI) for healthy ($n = 14$), mild ($n = 15$), and moderate (n = 10) COVID-19 monocytes stimulated with LPS (mean±s.e.m.). For **d** and **f**, gene expression values are scaled by row; red indicates relatively high expression, and blue low expression. Both rows and columns are clustered using Euclidean distance and Ward's method. Mixed model with Tukey's post hoc test for **g** and **i**. One-way ANOVA with Tukey's test for **h**. For **g** and **i**, two-way ANOVA with Tukey's correction for baseline vs. other time points within the same group. *$p < 0.05$, ***$p < 0.001$ for healthy individuals, #$p < 0.05$, ##$p < 0.005$ for mild COVID-19 patients, $$$p < 0.001 for moderate COVID-19 patients. ****$p < 0.0001$. Source data are provided as a Source Data file.

We subsequently took advantage of the available clinical data of the patients for which RNA-seq analysis of monocytes was performed. For these patients we had information of the plasma concentration of D-dimer at the time of blood collection. D-dimer is routinely used in the clinic as a biomarker for activation of the coagulation and fibrinolysis systems[49] and it has been extensively investigated for the diagnosis, monitoring and treatment of venous thromboembolic diseases, for which it is used routinely[50]. Plasma D-dimer concentration positively correlates with venous thromboembolic diseases in the general population and in patients with COVID-19[51]. We decided to group samples into high and low plasma D-dimer concentration as a readout of potential thrombotic issues. While most patients whose monocytes were used for RNA-seq had elevated D-dimer concentrations at the time of blood collection (>500 ng/ml, Dataset 9), we could clearly identify two groups of patients: those with D-dimer concentration >1600 ng/ml (High D-dimer, range 1647–20,000 ng/ml) and those with D-dimer concentration <1000 ng/ml (Low D-dimer, range 966–534 ng/ml). We then examined the expression of genes belonging to the pathways potentially associated with hemostasis and coagulation in both ex vivo RNA-seq data and RNA-seq data from stimulated monocytes in healthy individuals from low and high D-dimer COVID-19 groups (Fig. 6d–f).

Using the RNA-seq datasets from ex vivo isolated monocytes, we examined the expression of those genes that were significantly enriched in two pathways: "Hemostasis" and "Platelet activation, signaling and aggregation" (genes and pathways can be found in Dataset 4). In agreement with the increased capacity to form MPAs, we observed an increased expression of genes associated with these pathways in a D-dimer concentration-dependent manner (Fig. 6d). Moreover, while not all genes were significantly upregulated in the high D-dimer group, 10 out of 12 showed a trend toward an increased gene expression as D-dimer concentration increased (Fig. 6e). This observation was specific for these two hemostasis-related pathways, as the trend was not observed when we examined the normalized read counts of those genes enriched in another significantly enriched pathway i.e., "Transmembrane transport of small molecules", with only 5 out of the 14 enriched genes showing a trend towards a D-dimer concentration-dependent increase in expression (Supplementary Fig. 12).

We also examined the expression (normalized read counts) of those genes enriched in pathways potentially related to a pro-thrombotic signature in RNA-samples from COVID-19 monocytes stimulated with UV-inactivated SARS-CoV-2, again dividing the samples into low and high D-dimer concentrations in plasma (Fig. 6f). The pathways tested were "Integrin cell surface interactions", "Extracellular matrix organization", "Response to elevated platelet cytosolic $Ca^{2+}$", "Signaling by PDGF", "Hemostasis" and "Platelet aggregation (plug formation)" (Genes and pathways can be found in Dataset 7; 81 genes in total). Interestingly, while all the genes were significantly upregulated

in COVID-19 patients as compared to healthy controls, no differences in their expression were found between the two groups of patients with low and high plasma D-dimer concentration in a heatmap built with z-score transformed normalized gene counts of all the genes within the abovementioned pathways (Fig. 6f). Moreover, heatmaps of individual pathways did not reveal any differences in the expression of the corresponding genes based on plasma D-dimer concentration (Supplementary Fig. 13). This observation suggests that secondary SARS-CoV-2 sensing may switch monocyte phenotype and functionality in a D-dimer concentration-independent manner.

**Endotoxin tolerance signature enriched in COVID-19 monocytes**

A number of works have suggested similarities between the characteristics of the immune response in COVID-19 patients and those of septic individuals, including multiple organ dysfunction, immunosuppression, coagulopathies and acute respiratory failure[52]. To determine the similarities between the transcriptional signature of COVID-19 monocytes with that of sepsis monocytes, we utilized publicly available microarray gene expression data on sepsis monocytes and healthy controls[53] and we tested the estimated fold changes for correlation with those from our ex vivo (Fig. 7a) and activated (Fig. 7b) COVID-19 and healthy monocytes. The sepsis dataset compared 6 age-matched healthy individuals to 8 adult Gram-negative urinary sepsis patients, with samples taken within 4 h of hospital admission[53]. No clear correlation was observed in any of the two contrasts, which suggests that the transcriptional signature of CD14+ monocytes in moderate COVID-19 is not similar to that of monocytes in sepsis.

The lack of cytokine expression, activation of costimulatory receptors, impaired antigen presentation potential and metabolic impairments displayed by moderate COVID-19 monocytes resembled the phenotype observed in LPS-induced tolerance[54]. We have previously defined an endotoxin tolerance gene expression signature from publicly available microarray data on monocytes stimulated in vitro with LPS[55] that comprises 398 genes. Out of these, 318 genes were detected in our RNA-seq dataset. We tested for correlation of the endotoxin tolerance signature with ex vivo (Fig. 7c) and activated (Fig. 7d) COVID-19 monocytes, and while ex vivo COVID-19 monocytes did not display a clear correlation with the tolerance signature, activated COVID-19 monocytes displayed similar directionality of expression in those genes from the tolerance signature that were detected in the dataset. These data were further confirmed in barcode plots (Fig. 7e), showing a statistically significant enrichment of the endotoxin tolerance gene signature in the list of differentially expressed genes from stimulated COVID-19 monocytes compared to healthy controls, for both upregulated and downregulated genes. These data are in agreement with the observed diminished response of COVID-19 monocytes to secondary LPS stimulation in both total PBMC (Fig. 4c, f) and isolated CD14+ monocytes (Supplementary Fig. 14).

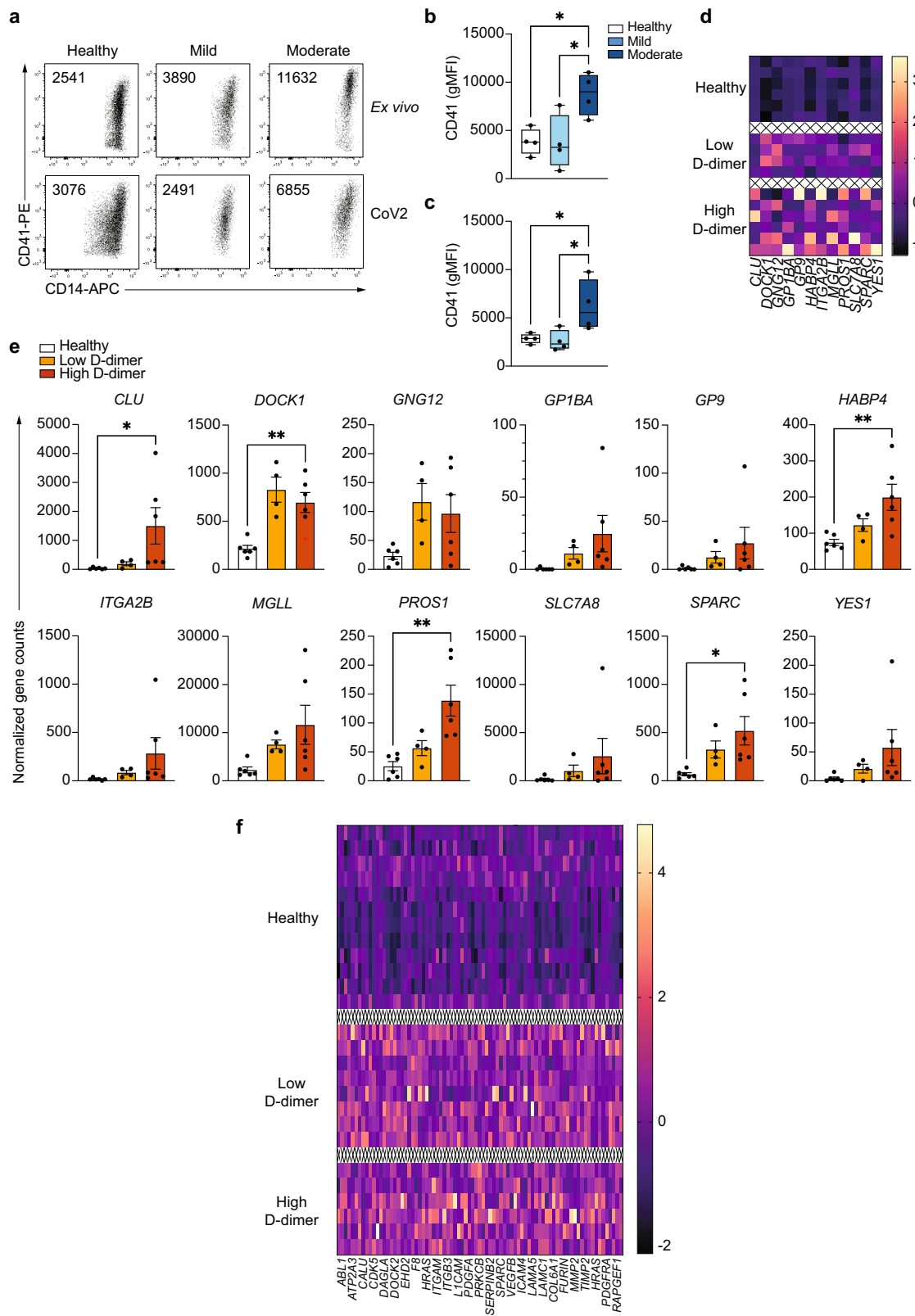

## Discussion

Here we employed metabolic, transcriptomic and functional assays to identify a number of phenotypic and functional alterations in COVID-19 monocytes that characterize moderate disease and we have provided the functional characteristics of monocyte responses in mild SARS-CoV-2 infections as an example of an efficiently and successfully cleared infection without excessive immunopathology. Important alterations in metabolism and transcriptional signatures characterize moderate COVID-19 monocytes and are important aspects of a global unresponsiveness phenotype upon pathogen sensing characterized by a transcriptional switch from canonical innate immune functions to a pro-thrombotic signature. This pro-thrombotic phenotype was further confirmed functionally, and is in agreement with clinical observations that patients with moderate and severe COVID-19 are at higher risk of

**Fig. 6 | Monocytes from moderate COVID-19 patients are functionally pro-thrombotic.** Monocytes were isolated from healthy individuals, mild and moderate COVID-19 patients, and left unstimulated or stimulated with UV-inactivated SARS-CoV-2 for 20 h. **a** Representative dot plots of the expression of CD41 on ex vivo isolated (upper row) or stimulated (lower row) monocytes from healthy individuals (left), mild (middle), and moderate (right) COVID-19 patients ($n = 4$ individuals per group) after co-culture with freshly isolated platelets from a healthy individual. Numbers in each dot plot represent CD41 gMFI. **b, c** Summary of CD41 gMFI on unstimulated (**b**) or stimulated (**c**) monocytes after co-culture with healthy donor platelets ($n = 4$ individuals in each group). Boxes in **b** and **c** extend from the 25th to the 75th percentiles and whiskers extend down to the minimum and up to the maximum values. **d** RNA-seq datasets from ex vivo isolated monocytes from moderate COVID-19 patients were grouped into low and high D-dimer concentrations. Heatmap of z-score-transformed normalized read counts of significantly upregulated genes in "hemostasis" and "platelet activation, signaling, and

aggregation" pathways in healthy ($n = 6$), low D-dimer concentration ($n = 4$) and high D-dimer concentration ($n = 6$) moderate COVID-19 monocytes. Gene expression values are scaled by column, and each row represents one individual. **e** Summary of normalized gene counts of the genes in **d**, shown as mean±s.e.m. **f** RNA-seq datasets from stimulated monocytes from moderate COVID-19 patients were grouped into low and high D-dimer concentrations. Heatmap of z-score-transformed normalized read counts of significantly upregulated genes in "Integrin cell surface interactions", "Extracellular matrix organization", "Response to elevated platelet cytosolic $Ca^{2+}$", "Signaling by PDGF", "Hemostasis" and "Platelet aggregation (plug formation)" pathways in healthy ($n = 12$), low D-dimer concentration ($n = 8$) and high D-dimer concentration ($n = 6$) moderate COVID-19 monocytes. Gene expression values are scaled by column, and each row represents one individual. One-way ANOVA with Tukey's correction for multiple comparisons for **b, c, e**. *$p < 0.05$, **$p < 0.005$. Source data are provided as a Source Data file.

developing hemostasis issues[42,43,56]. The initial inflammatory response mounted upon SARS-CoV-2 infection could potentially drive the changes in monocyte functionality, as inflammation is well known to activate the coagulation system[57–59]. Moreover, while our results are focused on the functionality of monocytes, previous data have shown that platelets from patients with COVID-19 are activated ex vivo during the acute phase of disease and have increased capacity to form monocyte-platelet aggregates[48], which supports the notion of inflammation driving the initial activation and functional switch of these cell types, promoting the initiation of hemostasis issues and potentially the diminished innate immune functions upon secondary stimulation. In addition, the metabolic defects observed in COVID-19 monocytes probably underlie the observed diminished response to secondary stimulation, as they modulate innate immune functions including cytokine expression, activation, phagocytic capacity, etc[30,60,61]. More mechanistic studies are needed to understand the link between coagulation and hyporesponsiveness to secondary monocyte stimulation in COVID-19 patients.

A question that remains to be answered is the driver(s) of the described circulating monocyte dysfunction. Ex vivo isolated monocytes from moderate COVID-19 patients are pro-thrombotic while maintaining some innate immune functions (Fig. 2). However, secondary pathogen sensing ex vivo triggers a switch in COVID-19 monocyte gene expression signature and functionality from canonical innate immune functions to pro-thrombotic phenotype. It remains to be determined whether any soluble factors in the microenvironment contribute to this reprogramming, or even the direct infection of monocytes by SARS-CoV-2, which has been previously suggested[62]. The phenotype we observe in circulating monocytes is in clear contrast with the functionality of monocyte-derived macrophages in the lung of COVID-19 patients[10]. In this regard, our study is limited by the lack of bronchoalveolar lavage fluid (BALF) paired samples to compare the phenotype and function of circulating monocytes with those infiltrating the target tissue. However, some previous publications examining paired airway and blood samples have shown differences in the signatures of circulating and lung innate immune cells, with low HLA-DR expressing, dysfunctional monocytes in the blood, and hyperactive airway monocyte and macrophages producing pro-inflammatory cytokines[10,29,63]. The underlying mechanisms for these differences remain elusive. During the course of viral infections, circulating monocytes rapidly leave the bloodstream and migrate to target tissues, where, after pathogen sensing and/or other microenvironmental stimuli, they differentiate into macrophages and/or dendritic cells. In this study, we examined the functionality of monocytes during the acute phase of disease, early after symptom onset. It remains to be determined whether these dysfunctional monocytes have the capacity to migrate to the lungs and contribute to lung inflammation, or whether their dysfunction is such that migration is impaired and monocyte migration only occurred during the very initial phases of infection

before monocyte acquired the impairments observed in this study. Of note, some of the defective pathways displayed by COVID-19 monocytes, as for example glycolysis, have been shown to be essential for migration of other cells to target tissue[64,65]. Finally, the results described in this study beg the question of whether the functional impairments observed in monocytes during the acute phase of infection are COVID-19-specific. While stimulation with other viruses and bacterial products led to similar altered immune phenotypes in COVID-19 monocytes (Fig. 4), it seems likely that these processes occur with other moderate respiratory viral infections, as has been shown in seasonal Influenza vaccination[66]. Longitudinal studies of monocyte dynamics during SARS-CoV-2 and other respiratory viral infections using both blood and BALF samples are warranted to answer these questions.

## Methods

### Participants and clinical data collection

Disease severity was categorized based on the WHO ordinal classification of clinical improvement, where 0 (uninfected) describes people with no clinical or virological evidence of infection, 1-2 describe ambulatory patients without (1) or with (2) limitation of activities, and 3-4 corresponds to hospitalized patients with no oxygen therapy (3) or oxygen by mask or nasal prongs (4). Peripheral blood was collected from all participants and processed following a common standard operating protocol. For inpatients, clinical data were abstracted from the electronic medical records into summary participant sheets. Participant group characteristics are summarized in Dataset 1.

Healthy donors (WHO 0) were Imperial College staff with no prior diagnosis of or recent symptoms consistent with COVID-19, and where possible, were matched in age and sex distribution with COVID-19 patients. None of the participants of this study were COVID-19 vaccinated.

Blood samples from the COVID-19 patients examined in this work come from two different studies. COVIDITY study is a prospective observational serial sampling study of whole blood to observe the evolution of SARS-CoV-2 infection to characterize the host response to infection over time in peripheral blood (ethics approval obtained from the Health Research Authority, South Central Oxford C Research Ethics Committee). The population of study were >18-year-old patients and/or staff at Imperial College Healthcare NHS Trust/Imperial College London with confirmed COVID-19 from a positive SARS-CoV-2 RT-PCR test from NHS laboratories or Public Health England. After informed consent was obtained, samples were taken 3-14 days after symptom initiation and were classified as 1 or 2 disease severity.

Samples from patients with moderate COVID-19 admitted to hospitals in London (Hammersmith Hospital, Charing Cross Hospital, Saint Mary's Hospital) and eligible to participate in the MATIS trial (NCT04581954)[67] provided consent (ethics approval by the Health

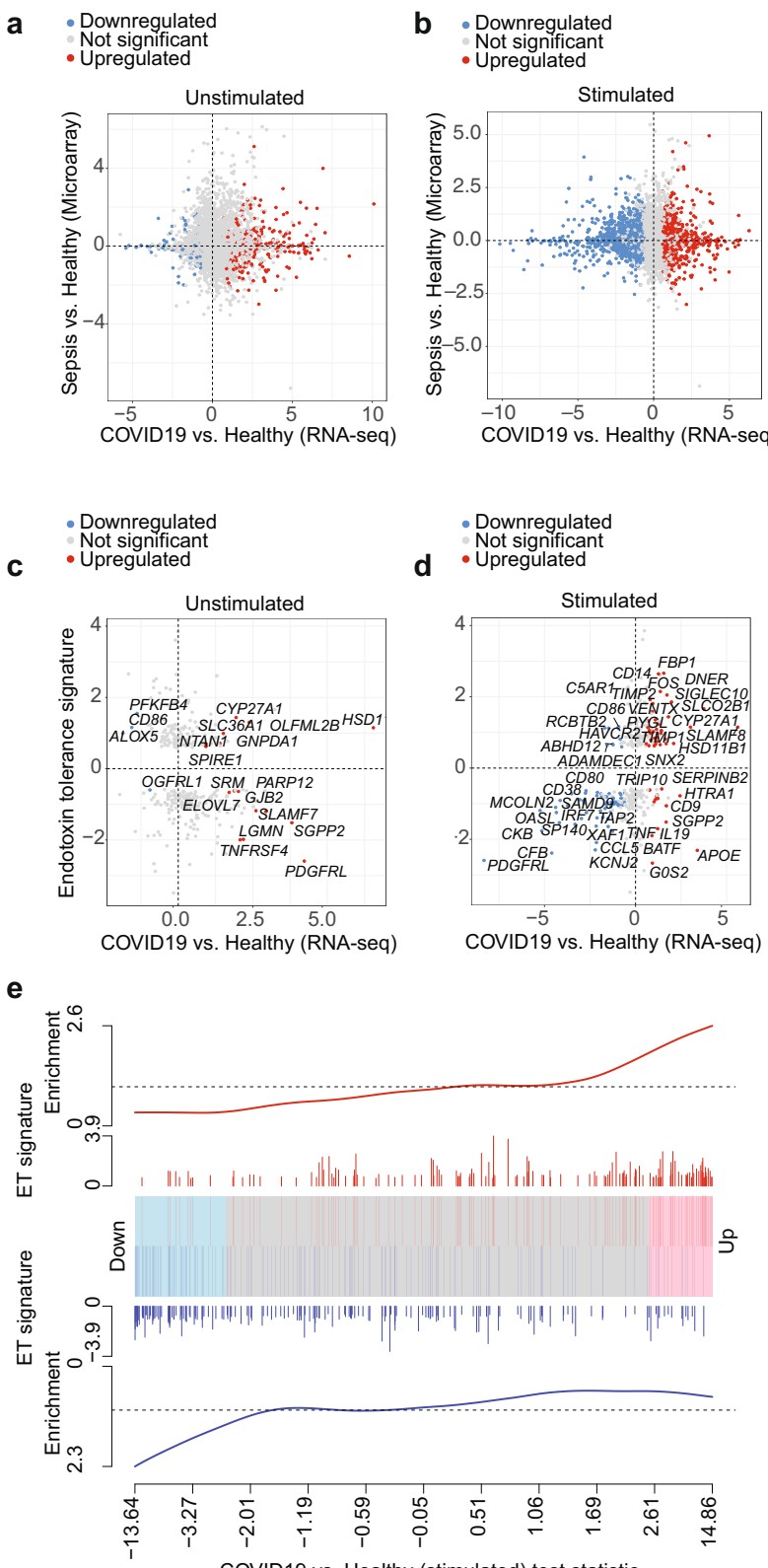

Research Authority, London-Surrey Borders Research Ethics Committee) and blood was collected 3-14 days after disease onset and 0-2 days after hospitalization and positive PCR, and before study treatment initiation. Moderate patients displayed mild or moderate COVID-19 pneumonia, defined as grade 3 or 4 WHO severity. Samples were collected from March 2020 to February 2021 and none of the participants had received a COVID-19 vaccine.

**PBMC isolation, storage, and thawing**

Peripheral blood mononuclear cells (PBMCs) were isolated by Ficoll Hypaque (GE Healthcare) gradient centrifugation <4 h after blood collection. The PBMC layer was collected, washed with PBS, resuspended at 20 million cells/ml in fetal bovine serum supplemented with 10% DMSO and stored at −150 °C or liquid nitrogen. For PBMC thawing, vials were thawed in a pre-warmed water bath at 37 °C and immediately

**Fig. 7 | Endotoxin-induced tolerance signature significantly enriched in COVID-19 monocytes.** Correlation plot of sepsis vs. healthy individual gene expression signature and ex vivo (**a**) or stimulated (**b**) COVID-19 vs. healthy individual monocyte gene expression signature. Each point represents a gene detected in both the sepsis public microarray dataset and the COVID-19 RNA-seq dataset. The $\log_2(FC)$ between sepsis and healthy controls is plotted against the $\log_2(FC)$ for ex vivo COVID-19 monocytes vs. healthy control monocytes, and the points are colored according to the significance and direction of effect in the COVID-19 contrast (grey, not significant; red, significantly upregulated; blue, significantly downregulated). Correlation plot of endotoxin-induced tolerance gene signature and ex vivo (**c**) or stimulated (**d**) COVID-19 vs. healthy monocyte signature. Each point represents a gene detected in both the endotoxin gene signature public dataset and our COVID-19 vs. healthy RNA-seq dataset. The $\log_2(FC)$ between endotoxin tolerance and LPS response is plotted against the $\log_2(FC)$ for COVID-19 vs. healthy monocytes, and the points colored according to the significance and direction of effect in the COVID-19 contrast as in **a**. Some of the most differentially expressed genes in the COVID-19 vs. healthy monocyte dataset are identified in the plot. **e** Barcode plot showing enrichment of the endotoxin tolerance (ET) gene set in the differential gene expression results for SARS-CoV-2-stimulated COVID-19 vs healthy monocytes. The ranked test statistics from DESeq2 for the SARS-CoV-2-stimulated COVID-19 vs. healthy contrast are represented by the central shaded bar, with genes downregulated in COVID-19 on the left and upregulated genes on the right. The ranks of the ET gene set within the COVID-19 contrast are indicated by the vertical lines in the central bar. The weights of these genes ($\log_2(FC)$ from the ET gene expression analysis) are indicated by the height of the red and blue lines above and below the central bar. The red and blue lines at the top and bottom indicate relative enrichment of the ET genes (split into genes with positive and negative FCs in the ET contrast) in each part of the plot.

transferred to a 15 ml conical tube where 5 ml of warm complete media (37 °C) were added drop by drop. The tubes were subsequently centrifuged for 10 min at 250 x g and resuspended in warm media. The media used was RPMI 1640 supplemented with 2 nM L-glutamine, 5 mM HEPES, and 100 U/µg/ml penicillin/streptomycin (Biowhittaker, Walkersville, MD), 0.5 mM sodium pyruvate, 0.05 mM non-essential amino acids (Life Technologies, Rockville, MD), and 5% human AB serum (Gemini Bio-Products, Woodland, CA).

**Flow cytometry staining for monocyte immunophenotyping**
PBMCs were thawed and rested for 2 h at 37 °C in RPMI 1640 media supplemented with 2 mM L-glutamine, 5% human AB serum, and 1x Penicillin and Streptomycin. For ex vivo phenotypic characterization, 300,000-500,000 PBMC were stained with LIVE/DEAD Fixable Dead Cell Dyes (Thermo Fisher Scientific) according to the manufacturer's specifications. A Fc receptor (FcR) blocking step was performed using FcR Blocking Reagent Human (Miltenyi Biotec) before cell surface antibody staining (Supplementary Table 1). The antibodies used in the stainings were the following: CD14 (61D3, eBioscience), CD3 (UCHT1, BD), CD19 (HIB19, BD), CD1c (L161, Biolegend), CD40 (5C3, Biolegend), CD141 (M80, Biolegend), CD304 (12C2, Biolegend), CD86 (BU63, Biolegend), CD80 (BB1, BD Pharmigen), HLA-DR (L243, Biolegend), CD301 (H037G3, Biolegend), HLA-ABC (W6/32, Biolegend), TIM-3 (F38-2E2, Invitrogen), PD-1 (EH12.2H7, Biolegend), and CD16 (3G8, BD). Cells were subsequently fixed using the Foxp3 staining buffer kit (Thermo Fisher Scientific) following the manufacturer's recommendations and resuspended in 250 µl of PBS. Classical monocytes were gated on size and granularity, live cells, exclusion of other main cell lineages (CD3⁻ CD56⁻CD66b⁻CD16⁻CD19⁻) and CD14⁺ (Supplementary Figure 15).

For intracellular staining, the aforementioned protocol was used, and an additional step for intracellular staining was added after fixation. The antibodies used for intracellular staining were the following: TNF (Mab11, Biolegend) and IL-10 (JES3-907, Thermo Fisher Scientific). Intracellular staining was performed using the Foxp3 staining buffer kit.

Samples were run on a Fortessa instrument (BD Biosciences) and analyzed using FlowJo v.10.

Dimensionality reduction and tSNE plots were obtained by downsampling each of the 15 samples per group (healthy, mild COVID-19 and moderate COVID-19) to 1,500 monocytes per sample, and the concatenated sample was used to calculate tSNE axes using 1,000 iterations, perplexity of 40 and the default learning rate (4734). In order to obtain cell clusters, we used Phenograph[68] plugin in FlowJo, with k = 166 and all compensated parameters.

**Generation of virus stocks**
SARS-CoV-2 virus (SARS-CoV-2/England/IC19/2020 isolate, kindly provided by Wendy S Barclay) was expanded in Vero-E6 cells. Briefly, Vero-E6 cells were plated in serum-free medium (OptiPRO SFM containing 2x GlutaMAX) in T75 flasks and infected with SARS-CoV-2 at a multiplicity of infection of 0.1 and a final volume of 5 ml. Cells were

incubated for 2 h at 37 °C, 5% $CO_2$, after which the inoculum was removed and complete medium without serum was added to the culture. Cells were incubated for 3–5 days (until cytopathic effects were observed). Subsequently, cell culture supernatant was collected, centrifuged at 1000 x g, 4 °C for 15 min and transferred to a new 50 ml tube for a second centrifugation at 1000 x g, 4 °C for 15 min. Viral supernatant was collected, filtered through 0.45 µm and an aliquot was taken for titration. The rest of the supernatant was UV-inactivated and concentrated using Retro-X concentrator (Takara Bio), following manufacturer's recommendations and published protocols[69,70].

Human coronaviruses (CCCoV) 229E, OC43 and NL63 strains (Public Health England) were expanded in MRC-5 (kindly provided by Dr Rob White, Imperial College London), BSC-1 (Public Health England) and LLCMK2 (Public Health England), respectively. Briefly, cell lines were plated in serum-free medium (DMEM, 1x non-essential amino acids) in T75 flasks and infected with CCCoV (229E, OC43 or NL63) at a multiplicity of infection of 0.1 and a final volume of 5 ml. Cells were incubated for 2 h at 37 °C, 5% $CO_2$, after which the inoculum was removed and medium without serum was added to the culture. Cells were incubated for 3-5 days (until cytopathic effects were observed). Subsequently, cell culture supernatant was collected, centrifuged at 1000 xg, 4 °C for 15 min and transferred to a new 50 ml tube for a second centrifugation at 1000 xg, 4 °C for 15 min. Viral supernatant was collected, filtered through 0.45 µm and an aliquot was taken for titration. The rest of the supernatant was heat-inactivated and concentrated using Retro-X concentrator (Takara Bio), following manufacturer's recommendations and published protocols[69,70].

**Titration of virus stocks**
For SARS-CoV-2 titration, samples were serially diluted in OptiPRO SFM, 2X GlutaMAX (1:10) and added to Vero cell monolayers for 1 hour at 37 °C, 5% $CO_2$. The inoculum was subsequently removed and cells were overlayed with DMEM containing 0.2% w/v bovine serum albumin, 0.16% w/v $NaHCO_3$, 10 mM HEPES, 2 mM L-Gutamine, 1X P/S and 0.6% w/v agarose. Plates were incubated at 37 °C, 5% $CO_2$ for 3 days. The overlay was then removed and monolayers were stained with Crystal violet solution for 1 hour at room temperature. The plates were washed with water and dried, and the virus plaques were then counted.

For CCCoV titration, viral supernatants were serially diluted in DMEM, non essential amino acids (1:10) and added to MRC-5 (229E strain), BSC-1 (OC43 strain) or LLCMK2 (NL63 strain) cell monolayers for 1 hour at 37 °C, 5% $CO_2$. The inoculum was subsequently removed and cells were overlayed with DMEM medium for 4-5 days (until cytopathic effects were observed). An endpoint dilution assay was used to determine viral infectivity titers[69].

**Ex vivo stimulation assays**
PBMC were thawed and rested for 2 h at 37 °C in complete media. 250,000 PBMC were plated in polystyrene plates (Corning) to prevent unspecific stimulation of monocytes by adherence to the plastic

plate[71]. Cells were stimulated with vehicle, UV-inactivated SARS-CoV-2 (CoV-2), 100 ng/ml LPS or a mixture of heat-inactivated common cold coronaviruses consisting of the 229E, OC43, and NL63 strains (CCCoV) at $10^6$ viral particles per $10^6$ cells for 20 h. For intracellular stainings, GolgiStop™ (BD Biosciences) was added to the cultures 10 h after stimulation for a total of 10 h. For ex vivo stimulation of monocytes, $CD14^+$ monocytes were isolated using a positive selection magnetic sorting kit (StemCell Technologies, UK) from total PBMC and stimulated following the same protocol as for total PBMC.

## RNA isolation and sample preparation for RNA-seq analysis

$CD14^+$ monocytes were sorted in a FACS Aria from total PBMC either ex vivo or after a 20-hour stimulation with $10^6$ UV-inactivated SARS-CoV-2 viral particles per $10^6$ cells and lysed with RLT Plus buffer (Qiagen). The following gating strategy was used to sort $CD14^+$ monocytes: size and granularity-based gating on FSC vs SSC, live cells (cells negative for propidium iodide), $CD3^-CD19^-CD66b^-CD16^-CD56^-CD14^+$. Post-sort purity was determined in all samples by flow cytometry (by CD14 staining) and was above 98% in all samples. RNA was isolated using the RNeasy Micro Plus Kit (Qiagen) following the manufacturer's guidelines in Appendix D of the Qiagen RNeasy handbook. RNA quality was quantified using the Agilent RNA 6000 Pico Kit (Agilent Technologies) following the manufacturer's guidelines. RNA samples were stored at −80 °C until further processing.

## Monocyte-platelet aggregate (MPA) assay

Frozen PBMC were thawed and rested for 2 h before $CD14^+$ isolation using a positive selection kit (StemCell Technologies, UK). Isolated monocytes were stimulated or not with UV-inactivated SARS-CoV-2 as above. Platelets were isolated from platelet rich plasma (PRP) of healthy donor fresh blood right after collection into citrate tubes by centrifugation. Co-cultures were set at a monocyte:platelet ratio of 1:25 and incubated for 24 h at 37 °C, 5% $CO_2$. Subsequently, cells were washed with Hepes buffer and stained for flow cytometry with a viability dye as above and antibodies to CD14, CD41 and CD33. Gating strategy shown in Supplementary Fig. 15.

## RNA-seq analysis

RNA-sequencing was performed by the Oxford Genomics Centre. PolyA-enriched strand-specific libraries were prepared using NEBNext Ultra II Directional RNA Library Prep Kits (Illumina). All samples were pooled together and 150 bp PE reads were sequenced on a Novaseq system, resulting in a median read count of 28 million per sample.

Raw data was processed using the Sanger Nextflow RNA-seq pipeline. Briefly, reads were aligned to the reference genome (GRCh38.99) using STAR v2.7.3[72] in the two-pass mode (using ENCODE recommended parameters) and gene expression was quantified using featureCounts[73]. Mapping table and quality control metrics from FastQC and RNA-SeQC[74] indicated high data quality for all samples with no outliers detected.

RNA-seq data analysis was performed in R v4.1 in Rstudio Server. Features that did not have at least 10 reads in at least 6 samples (the size of the smallest biological subgroup) were filtered out using the genefilter package[75], resulting in a processed data set on 16,328 features. Principal component analysis (PCA) with the prcomp function was used to explore the relationship between samples, after the filtered gene counts were transformed using a regularized log transformation from the DESeq2[76] package.

Differential gene expression analysis was carried out using DESeq2, comparing unstimulated monocytes from COVID-19 patients ($n = 10$) to unstimulated monocytes from healthy controls ($n = 6$), and SARS-CoV-2-stimulated monocytes from COVID-19 patients ($n = 14$) to stimulated monocytes from healthy controls ($n = 12$). Genes with FDR < 0.05 and a fold change (FC) ≥ 1.5 were deemed significantly differentially expressed. Pathway enrichment analysis was performed

using Fisher's exact test in XGR[25] with annotations from Reactome, using all genes retained in the processed RNA-seq data as the background, and employing the xEnrichConciser options. An adjusted p-value (FDR) threshold of 0.05 was used to identify significantly enriched pathways. Pheatmap package was used to draw heatmaps illustrating variation in gene expression across samples.

For testing the enrichment of the sepsis signature in our datasets, publicly available microarray gene expression data on sepsis patients and healthy controls were accessed using GEOquery (GSE46955)[53]. Gene expression between patients and controls was compared using limma[77], for both the unstimulated and stimulated conditions. Subsequently, the estimated fold changes were tested for correlation with those from the COVID-19 vs. healthy control results. Where multiple probes were available for the same gene in the microarray dataset, the top-ranked probe was selected for the comparison.

For comparison to the endotoxin-induced tolerance signature, we have previously defined an endotoxin tolerance gene signature[78] from publicly available microarray data on in vitro LPS-stimulated monocytes. Briefly, two datasets (GSE15219[55] and GSE22248[79]) were accessed through GEO. Genes that were differentially expressed following a single LPS treatment (LPS response genes), and that were also differentially expressed between singly- and doubly-stimulated cells were identified. This resulted in an endotoxin tolerance gene signature comprising 398 genes, of which 318 were detected in the RNA-seq dataset. We tested for enrichment of this gene set in the COVID-19 versus healthy contrasts using the geneSetTest function and barcodeplot functions from limma.

## Quantification of mRNA expression by real-time PCR

Isolated RNA was converted to complementary DNA by reverse transcription (RT) with random hexamers and Multiscribe RT (TaqMan Reverse Transcription Reagents; Thermo Fisher Scientific). For *IFITM2* expression assays, the Hs00829485_sH probe was used (Thermo Fisher Scientific). The reactions were set up using the manufacturer's guidelines and run on a StepOnePlue Real-Time PCR Machine (Thermo Fisher Scientific). Values are represented as the difference in cycle threshold (Ct) values normalised to *GAPDH* expression (Hs02786624_g1) for each sample as per the following formula: Relative RNA expression = $(2\text{-}\Delta Ct) \times 1000$[80].

## Metabolic profiling using SCENITH™

SCENITH™ is a flow cytometry-based method for profiling energy metabolism with single cell resolution[31] ex vivo or after in vitro stimulation in sorted cells or complex cell mixtures. It uses puromycin incorporation to nascent proteins as a measurement for protein translation, which is tightly coupled to ATP production and therefore can be used as a readout for the energetic status of the cells at a given time.

PBMC were plated at 250,000–300,000 cells per well in 96 well plates and rested for 2 h at 37 °C, 5% $CO_2$ for ex vivo stainings, or rested for 2 h and stimulated for 20 h with 100 ng/ml LPS. Subsequently, cells were treated for 45 min at 37 °C, 5% $CO_2$ with Control (vehicle, Co), 100 mM 2-deoxy-D-glucose (DG, Sigma-Aldrich), 1 μM oligomycin (O, Sigma-Aldrich) or a combination of both drugs (DGO). 10 μg/ml puromycin was added to all conditions for the same amount of time. Cells were subsequently washed with room temperature PBS and stained for viability, cell surface markers and fixed as described above. Intracellular staining of puromycin was performed using the anti-puromycin monoclonal antibody (1:600 dilution, clone R4743L-E8) for 45 min at 4 °C. The SCENITH™ kit containing stabilized puromycin, anti-puromycin antibody and metabolic inhibitors were kindly provided by Dr Argüello. 

For the analysis of the energetic status of cells, puromycin geometric mean fluorescence intensity was analyzed in each of the four abovementioned conditions (Co, DG, O, DGO). To calculate the

percentage of glucose dependence, the following formula was used: 100*(Co-DG)/(Co-DGO). Mitochondrial dependence (%) was calculated as 100*(Co-O)/(Co-DGO). Glycolytic capacity (%) was calculated as 100-Mitochondrial dependence. Fatty acid and amino acid oxidation capacity (%) was calculated as 100-Glucose dependence.

## Metabolic profiling using Seahorse

Sorted CD14$^+$ monocytes from unstimulated or SARS-CoV-2-stimulated (20 h at 37 °C, 5% $CO_2$) PBMC were plated at a range of 80,000–120,000 in duplicates for healthy and COVID-19 sample pairs, based on the minimum cell number obtained for each pair of samples in individual experiments. An XFp real-time ATP rate assay kit (Agilent Technologies) was used following manufacturer's recommendations and samples were run in a Seahorse XF HS Mini Analyzer (Agilent Technologies). For basal oxygen consumption rate (OCR) and extra-cellular acidification rate (ECAR) measurements, 10 cycles were run and their average was taken as basal values per subject tested.

## Phosphoflow assays

For ex vivo phosphorylation assays, thawed PBMC were plated at 250,000 cells per well in 96 well polypropylene plates and rested for 2 h at 37 °C, 5% $CO_2$. PBMC were fixed with pre-warmed (37 °C) Cytofix (BD Biosciences) for 20 min at 37 °C, 5% $CO_2$ and permeabilized with Perm III buffer (BD Biosciences) overnight at −20 °C. Cultures were subsequently stained with CD3 (UCHT1, BD Biosciences), CD20 (H1, BD Biosciences), CD14 (M5E2, Biolegend), CD16 (B73.1, BD Biosciences), phospho-IRF3 (Ser 396, Bioss), phospho-NFkB p65 (Ser 529, BD Biosciences) in PBS for 1 hour at room temperature, washed with PBS and resuspended in 250 µl PBS.

For phosphorylation assays after LPS stimulation, PBMC were plated as above and stimulated with 100 ng/ml LPS for a total of 1 hour. Samples were fixed at 0, 5, 15, 30, 45, and 60 min after LPS addition for 20 min at 37 °C, 5% $CO_2$, and stained as above.

## Statistical analyses

Data were analyzed using GraphPad Prism version 9.4. Normal distribution of the data was tested using the Anderson-Darling and D'Agostino and Pearson normality tests, or Shapiro-Wilk test for those datasets with a small number of replicates. Normally distributed data by at least one of the two tests were analyzed using one- or two-way ANOVA when comparing more than two groups of one or two independent variables, respectively. A two-tailed t-test was used to compare two groups. Data are presented as mean ± s.e.m. Where data are presented as box and whiskers, the boxes extend from the 25$^{th}$ to the 75$^{th}$ percentile and the whiskers are drawn down to the minimum and up to the maximum values. Horizontal lines within the boxes denote the median. $p$ values <0.05 were considered statistically significant.

## Reporting summary

Further information on research design is available in the Nature Portfolio Reporting Summary linked to this article.

# Data availability

RNA-seq datasets generated during the current study have been deposited in the European Genome-Phenome Archive under accession code EGAD00001009800. Reference genome used was GRCh38.99 was used as reference genome. Gene expression data from sepsis patients was obtained from the Gene Expression Omnibus (GSE46955). Microarray data from in vitro LPS-stimulated monocytes (endotoxin tolerance signature) were also obtained from the Gene Expression Omnibus (datasets GSE15219 and GSE2224879). The raw numbers for charts and graphs are available in the Source data file whenever possible. Source data are provided with this paper.

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

## Acknowledgements

We thank the participants who volunteered for this study and the clinical teams of the COVIDITY and MATIS studies for patient recruitment and blood collection. We thank Dr. Parisa Amjadi and Ms. Radhika Patel for their help with flow cytometry sorting. A.K.M. is a Wellcome Trust Ph.D. scholar. K.L.B. and E.E.D. are funded by the Wellcome Trust [108413/A/15/D]. For the purpose of Open Access, the author has applied a CC BY public copyright license to any Author Accepted Manuscript version arising from this submission. We thank the Wellcome Sanger Institute's Human Genetics Informatics (HGI) team for mapping the RNA sequencing reads. This work was funded by a Rosetrees Trust grant (M971) to M.D.V.

## Author contributions

A.K.M. performed experiments, analyzed data and wrote the manuscript, K.L.B. analyzed the RNA sequencing data and wrote the manuscript, E.J. performed experiments, M.M.H.T. and R.C.S. performed experiments and analyzed data, C.P. and E.T. gathered clinical information from COVID-19 patients, LB prepared SARS-CoV-2 virus stocks, C.S. and N.G. performed experiments, C.E.S. and R.Q. provided patient samples, R.A. provided the SCENITH™ kit reagents and advised on SCENITH™ data analysis and interpretation, WSB provided SARS-CoV-2 virus stock, N.C. provided patient samples and advised on the clinical aspects of COVID-19, GPT provided COVID-19 patient samples and advised on the clinical aspects of COVID-19, EED supervised RNA-seq data analysis and wrote the manuscript, MDV designed the study, performed experiments, analyzed data, wrote the manuscript and obtained funding. All authors revised and contributed to the editing of the manuscript.

## Competing interests

The authors declare no competing interests.
