## [Peer Review File · Nature Communications]

REVIEWER COMMENTS

Reviewer #1 (expertise in monocyte biology, inflammation):

Mahler et al have performed a deep analysis of CD14+ monocytes from HV and COVID-19 patients with mild and moderate active disease.

They used multi-parameter FACS and mRNA-seq of purified unstimulated and SARS-CoV-2-stimulated CD14+ monocytes as screening assay. Some of the observed changes were confirmed with independent methods.

The strength of the study is that a large number of parameters were analyzed and thus also a large number of changes were found. The weakness of the study is that none of the interesting changes were confirmed in an independent second cohort of patients. Confirmation in a second cohort of patients may be less relevant, if the number of n is large (e.g. in Fig. 3) or if the results were confirmed with independent methods. Overall the study is (of course) descriptive even if some of the findings seem to fit nicely to clinical experience with COVID-19 patients (e.g. prothrombotic state of COVID-19 patients).

Specific comments:

Fig. 1: Based on the analysis of 9 parameters (Fig. 1b) in 15 patients / group 16 different subsets of monocytes were identified. I do not doubt that the analysis was performed in a technically adequate manner; however, I really would like to see that the data are reproducible in a second cohort of patients / HV. Please also analyze whether the changes seen by FACS in Fig. 1 are also detectable by mRNA-seq performed in Fig. 2. If not, please explain why.

Fig. 2: RNA-Seq was performed with purified CD14+ monocytes. I did not find information how the CD14+ monocytes were purified and how pure they really were. Contamination by small subsets of other cells (e.g. pDCs, neutrophils, or non-classical monocytes) may have a major impact on the mRNA-seq data.

Fig. 3: This figure is supposed to suggest that CD14+ monocytes from COVID-19 patients express less TNF α , CD40 and other molecules than CD14+ monocytes from HV upon in vitro challenge with SARS-CoV-2. If I understand the methods section correctly in vitro stimulation was performed with total PBMC and not with purified CD14+ monocytes. Therefore the differences found in monocytes could be secondary to a differential response of other cells (e.g. T cells, pDCs) to SARS-CoV-2. How would the data look like with purified CD14+ monocytes? Are the changes consistent with the mRNA-seq data of CD14+ monocytes stimulated with SARS-CoV-2?

Fig. 4: The main conclusion drawn by the authors from this mRNA-seq of SARS-CoV-2-stimulated CD14+ monocytes is a pro-thrombotic signature of monocytes from COVID-19 patients. The "pro-thrombotic signature" is also mentioned in the title of the manuscript. Thus I would like to see experimental data (e.g. in vitro assays for coagulation) that CD14+ monocytes from COVID-19 patients are prothrombotic if exposed to SARS-CoV-2. Just the interpretation of gene signature data is not sufficient to draw this conclusion. In addition, it would be interesting to know if monocytes from COVID-19 patients with clinically proven thrombosis have a more pronounced pro-thrombotic signature or are more pro-thrombotic in in-vitro assays

Fig. 5: Again only an interpretation of gene signature data. Please show that CD14+ monocytes from COVID-19 patients are less sensitive to LPS by comparing the LPS response of monocytes from COVID-19 patients and HV.

Reviewer #2 (expertise in innate immunity, monocytes and macrophages):

In this ms, Maher et al. analyse monocyte populations in mild and moderate COVID-19 patients. As compared to health donor monocytes, COVID-19 monocyte display:

1°) phenotypic modification at the level of cell surface proteins.

2°) decreased NF κ B activation but intact type I IFN response.

3°) impaired response to in vitro challenge with innate triggers associated to both defective NF κ B activation and impaired type I IFN responses and impaired glycolytic activity.

4°) the activation of a pro-thrombotic signature. The nature of the pro-thrombotic phenotype (id

est, genes supporting this conclusion) should be presented in the abstract section. This is important because most of these genes might have other functions as well.

The study is well performed and well presented. However, there is a consistent report of poorly conclusive analyses throughout the study.

The approach is essentially descriptive and do not provide any mechanistic on how this state of pro-thrombotic phenotype is promoted. A important flaw in design is to restrict the analysis to CD14+ monocytes, because this population could give rise to CD16+ monocytes as well. Therefore, a more thorough assessment of all monocytes populations could have been interesting.

Despite the identification of multiple epigenetic and metabolic phenotypes, no clear mechanistic frame/model is provided to explain the installation of this pro-thrombotic phenotype. As such, the study remains essentially descriptive. Interestingly, the phenotypic alterations seems more pronounced in moderate than mild COVID-19 donors.

Functionally, there is no proper assessment of the alleged pro-thrombotic activity of monocytes. Any functional assessment of the pro-thrombotic activity of SARS-Cov2 stimulated monocytes would increase greatly the impact of the study.

Altogether, despite its shortcomings, and despite the overstated claim that they have identified a mechanism, this study provides new and interesting insights on how monocytes are modified during mild/moderate COVID-19 infection. The findings reported here, are original and of interest for our understanding of COVID-19 pathogenesis.

As such, this study should pave the way for a further exploration of the role of monocytes in the dysregulation of coagulation associated to COVID-19.

Specific points of concern:

This sentence needs to be edited.

47. Moreover, contrasting

48. 51 observations regarding the development of cytokine storms vs. immunosuppression^{4,5} and the overactive or deficient type I IFN response in the lungs and in peripheral blood⁶⁻¹¹ have been 12 described for the role of myeloid cells in COVID-19

Figure 1

Line 104

"suggest an altered activation profile skewed towards an inhibitory phenotype".

Where mixed leukocyte reaction with allogenic T cells performed to support this hypothesis?

The number of samples in each clinical group should be indicated in the main text.

Cluster analysis: Like it is presented, this analysis is hard to read and interpret. The authors do not provide any statistical evidence than the cluster distribution change (or not) between clinical status group. The % of each cluster inside each group should be compared and the differences probed by appropriate statistical test. Otherwise, the pie chart are not informative.

In addition, differences are difficult to visualize given the high number of clusters used. Rand index could be calculated to probe the differences between groups.

Overall assessment of H3K27Ac, H3K4me3, H3K9me2, H3K27me3, alone, is poorly informative. It should be explained/clarified that this analysis is performed at the single cell level by FACS.

Figure 2

Figure 2C result is poorly reported and discussed in the text section which focuses on sup fig.

What is the metabolic prediction/biological meaning from the sphingolipid genes regulation?

The difference between 2C and 2h should be explained on the figure display to ease the reading.

Seahorse data should be brought back into the main figure to support the downregulation of glycolytic activity.

Figure 3

Does the poor response of mild COVID monocyte ex vivo translate into impaired T cell responsiveness (in a MLR assay)?

Figure 4

The authors report the identification of a pro-thrombotic signature by analysing SARS-Cov2 stimulated (ex vivo) monocytes.

Pathway analysis identifying "hemostasis and coagulation" should be brought back into the main figure.

As stated by the authors, these results are in agreement with reports of hypocoagulability in COVID

patients.

Functional assays assessing the pro-thrombotic activity of monocytes of ex vivo stimulated monocytes would further strengthen the message.

Reviewer #3 (expertise in macrophages, COVID-19, transcriptional regulation):

This manuscript describes an interesting ex vivo phenotype of classical monocytes from individuals infected with COVID-19. The authors identify differences in receptor expression, gene expression, modified histone expression and metabolic responses in ex vivo COVID-19 monocytes compared to controls. Additionally, they demonstrate that COVID-19 monocytes have muted secondary inflammatory responses when stimulated ex vivo and this is associated with differences in gene expression and metabolism.

Understanding mechanisms of immune dysfunction during COVID-19 infection are broadly interesting. This manuscript demonstrates an interesting phenotype in classical monocytes during COVID-19 infection, but concerns with the patient cohorts, lack of mechanistic data, and insufficient supporting data reduce my enthusiasm for the current version of the manuscript.

Primary comments:

1. Problematic interpretation when comparing mild and moderate COVID-19 cohorts:

There is insufficient descriptions of the 2 cohorts (mild and moderate COVID-19). Based on the information provided, the two cohorts appear poorly matched in terms of the time post symptom onset at which monocytes were collected. Previous profiling studies during viral infections (such as Dengue virus) and sepsis have showed time from symptom initiation is a strong predictor of peripheral blood responses. The average time from symptom initiation to sample collection for the mild group is almost 14 days. In immunocompetent individuals with mild disease we would expect the vast majority, if not all, of these individuals to have no recoverable virus and to be clinically well. Thus, the mild cohort is really an analysis of convalescence response to an acute COVID-19 infection? The moderate cohort samples were taken much earlier, soon after hospitalization, and at the point of worsening clinical disease.

- Given these monocytes are examined ex vivo, differences in monocyte responses between mild and moderate COVID-19 could reflect differences in monocytes after (mild) and during (moderate) COVID-19?

- Additional patient characteristics should also be provided: - include ranges, in addition to average, for each group as well as vaccination status (fully vaccinated, boosted, unvaccinated).

- The healthy cohort has N/A listed for all comorbidities, which suggests that anyone with any preexisting conditions (hypertension, diabetes, etc.) were excluded from this group. Is that correct? As expected, the moderate group has a significant number of comorbidities. Is it known if underlying comorbidities in the moderate group would impact the baseline expression of the monocyte markers evaluated and thus confound the analysis.

2. Lack of mechanism

The authors show differences in transcription, histone modifications and metabolism in monocytes, all of which have been shown following sepsis. Each of these effects could contribute to the poor response during ex vivo immune stimulation. More mechanistic insight into which mechanism is important in COVID-19 monocytes would be a great contribution that would elevate the importance of this manuscript.

3. Additional evidence is needed to completely support claims:

a) The authors claim "monocytes displayed defects in the epigenetic remodelling" which is supported by data that shows differences in total levels of modified histones. Although changes in levels may have implications for the underlying gene regulation, it's not necessarily clear what differences will actually mean. Location/locus dependent changes (i.e. ChIP) are much better correlated to relative changes in expression and should be provided to support differences in "epigenetic remodelling". Genome-wide ChIP-seq correlated to gene expression could provide support for claims of epigenetic remodeling as an important driver of changes in gene expression.

b) The authors claim that "transcriptionally, COVID-19 monocytes switched their gene expression signature from canonical innate immune functions to a pro-thrombotic phenotype". This claim is based on data from ex vivo monocytes that were further stimulated ex vivo (Fig 4). Given that this phenotype does not appear to be present in the analysis of ex vivo monocytes (Fig 2), it is not clear that this actually happens during COVID-19 infection in vivo.

- Clinically, the amount of SARS-CoV-2 circulating in the blood of moderately ill patients is

relatively low (or not detectable). What concentration of inactivated virus was used to stimulate the cells ex vivo? Is it comparable to in vivo?

- Additionally, the prothrombotic phenotype is based on the variable upregulation (Fig 4d) of a small subset of genes, many with functions outside of thrombosis (such as integrins, collagens and MMPs) (Fig 4b), in ex vivo monocytes stimulated with unclear quantities of inactivated virus. Based on this data, the claim of a prothrombotic expression signature is not well supported.

Other comments:

Figure 1:

Figure 1a – text states “monocytes from healthy individuals were clearly distinct from both mild and moderate COVID-19 on a tSNE plot” but legend states “a. tSNE plots obtained from a concatenated sample consisting of PBMC from n=15 healthy individuals, n=15 mild and n=15 moderate COVID-19 patients.”

- Is Figure 1a an analysis of classical monocytes or PBMCs? The specific cell types and markers analyzed to generate figure 1a should be explicitly stated in the text, legend or methods.

- The flow cytometry gating strategy with representative figures should be provided in supplementary materials

- Figure 1b – show all data points – (in addition to box-whisker or violin plot)

- Figure 1b – line 103 – The text suggests that PD1 is upregulated in moderate COVID-19 compared to control. However, PD1 expression in moderate is only up vs. mild COVID-19 but not vs. healthy control.

Figure 2:

- Figure 2B, Supp Fig 3, Supp Table 3: Text states that “Interestingly, pathway enrichment identified glycolysis as the most enriched pathway in COVID-19 monocytes together with metabolism of lipids and lipoproteins.” Although metabolism of lipids and lipoproteins is in these lists, it does not appear to be the 2nd most enriched (as this sentence suggests) in either supp fig 3 or supp table 3.

- Figure 2C is not called out in the text until after 2f but could be presumably called out prior to 2D.

- Figure 2e-g: Show all points (box-whisker vs violin plot) as well as representative flow plots for p-IRF3 and IκBα and pp65

- Figure 2f: uses an “expanded cohort” of COVID-19 patients. The clinical characteristics of the expanded cohorts should be described as part of supp table 1 or as an additional table. A brief description of what expanded means should be added to the text, is this just the addition of a few new individuals, or is this an entirely different cohort?

- Figure 2J and 2K: identify which cohort of individuals (initial, expanded or new) was utilized for these experiments and provide appropriate clinical details if not already done.

- please show individual data points in addition to bar plots

Figure 3:

- The text should specifically state that these experiments are performed with inactivated SARS-CoV-2 and hCoVs.

- add information to text or legend regarding the duration and magnitude (what quantity of virus/stimuli (LPS)) for each stimulation

Figure 4:

- state these experiments are performed with inactivated SARS-CoV-2

- add information to the text or legend regarding the duration and magnitude of inactive SARS-CoV-2 stimulation

- Figure 4a: Figure 2a shows that baseline gene expression is different in monocytes from COVID-19 and healthy controls before they are stimulated. How much of the variance observed is due to baseline differences rather than differences in the response to inactivated SARS-CoV-2 stimulation.

- Figure 4b: adding information to the volcano plot to show if movement to the left (-logFC) or right (+logFC) is representative of higher expression in control vs COVID would make the figure easier to interpret.

- Supp Fig 8 and Supp Table 6. Please clarify what 2 groups are being compared (inactive SARS-CoV-2 stimulated COVID-19 vs. healthy monocytes?) and what differentially regulated genes are used to generate these fig/tables. Is this both up and down regulated genes?

- Line 305 (in reference to Figure 4E). The text describes these genes as downregulated. Are they actually downregulated when compared to unstimulated? Or are they induced upon stimulation but to a lower degree than in healthy stimulated monocytes? Please adjust the text to make this distinction more clear.

Figure 5:

Figure 5a and 5b: characteristics of the sepsis cohort should be described (timing of when samples were drawn, type of infection, etc.) to provide important context for the reader.

Figure 5d – why use SARS-CoV-2 stimulation to compare with a tolerance signature that was specifically determined following LPS stimulation?

Text comments:

- Text lines 188-194. This is a long, complex sentence that could be broken into parts to enhance reader understanding. Given that severe COVID-19 has been associated with too much inflammatory immunopathology, it would be helpful to explain why lack of upregulation of cytokine expression important? This could be done in the discussion.

- Line 842 – mild of moderate should be mild of moderate

- Please provide details of how PBMCs were frozen and thawed in the methods as these could affect RNA expression.

- Each supplementary table should contain a title with clear information describing what is contained within that table. If it includes a differential expression list, then it should state what it is and what 2 groups are being compared to determine that list as well as the FC and FDR cut-off values. For example, the current titles “Table S7 - stim up pathway” is difficult to interpret. Does this mean what is up in stimulated vs unstimulated or is it moderate stimulated vs. healthy stimulated?

Point-by-point response to reviewers' comments.

Reviewer #1 (expertise in monocyte biology, inflammation):

- 1. Fig. 1: Based on the analysis of 9 parameters (Fig. 1b) in 15 patients / group 16 different subsets of monocytes were identified. I do not doubt that the analysis was performed in a technically adequate manner; however, I really would like to see that the data are reproducible in a second cohort of patients / HV. Please also analyze whether the changes seen by FACS in Fig. 1 are also detectable by mRNA-seq performed in Fig. 2. If not, please explain why.**

We thank the reviewer for this comment. We have examined the expression of the same 12 markers as in Figure 1 in a smaller cohort of patients with mild or moderate disease (due to limited availability of samples) and we have compared them to those of healthy individuals (n=5 participants per group, Revision Figure 1).

Revision Figure 1. **a.** tSNE plots obtained from a concatenated sample consisting of gated classical monocytes from total PBMC of healthy individuals, mild and moderate COVID-19 patients (n=5 individuals per group). **b.** Box and whiskers plots summarizing the median gMFI of the receptors analyzed (horizontal line inside boxes). The box extends from the 25th to the 75th percentile and the whiskers are drawn down to minimum and up to the maximum gMFI value obtained in each group. **c.** Number of cells per cluster identified by Phenograph. **d.** Heatmap of the expression of receptors per cell cluster displayed as modified z-scores using median values. One-way ANOVA with Tukey's correction for multiple comparisons for **b.** *p<0.05, **p<0.005, ***p<0.001.

Despite the smaller size of the cohort (and therefore the smaller number of cells that were analyzed in the concatenated sample, tSNE plots and clustering algorithms), we were still able to confirm that monocytes from healthy individuals, mild and moderate patients were clearly different based on the markers analyzed in the tSNE plot (Revision Figure 1A). Moreover, the expression of the cell surface markers examined in Figure 1 (HLA-DR, HLA-ABC, CD80, CD86, TIM3, CD14, PD1, CD301 and CD141) followed the same pattern of expression among clinical groups as in original Figure 1b. However, not all comparisons reached statistical significance, probably due to

the small number of samples analyzed per group (Revision Figure 1B). Clustering analysis following the same pipeline as for the initial cohort identified 15 different clusters of monocytes with the first 11 clusters containing approximately 91% of the total cells analyzed and with a comparable distribution of cell frequencies per cluster to Figure 1 (Revision Figure 1C). The difference in the number of clusters between the first (n=16 clusters) and the second cohort of patients (n=15 clusters) could be due to the smaller cell numbers (and smaller sample size in each group) analyzed from the second cohort (we analyzed 67635 cells in total in Figure 1 and only 21780 from the second cohort). Finally, heatmap representation of the clustering analysis also demonstrated that the main differences among clusters were mostly driven by the variability in the expression of the same markers as in Figure 1 (CD86, HLA-DR, HLA-ABC) with the bigger clusters in cell size having comparable phenotype to the ones in Figure 1. We have included a supplementary figure in the manuscript (Supplementary Figure 1) with a summary of these results (lines 114-116).

Regarding the mRNA-seq data, please see below the normalized read count values for the genes corresponding to the proteins examined in Figure 1 (Revision Figure 2). While most of the gene expression changes follow the same trend as their corresponding protein levels, not all of them are statistically significant, due in part to the complex translation regulation of some of these genes. Moreover, the HLA-DR antibody used in Figure 1 (clone L243) binds to a conformational epitope on HLA-DRA that depends on the correct folding of the $\alpha\beta$ heterodimer and thus, correlation with gene expression may not be straight forward. HLA-ABC antibody (clone W6/32) recognizes residues in the N-terminus of human β 2-microglobulin (*B2M*). Finally, the only marker that does not follow the same trend as its protein expression pattern is CD141 (*TBHD* gene), which is significantly upregulated in moderate COVID-19 patients compared to healthy individuals, while no differences in protein levels of CD141 are observed in the cohort of patients examined in Figure 1.

Revision Figure 2. Normalized read counts from CD14⁺ monocytes of healthy individuals (white bars) and moderate COVID-19 patients (blue bars) subjected to RNA-seq *ex vivo* after isolation (unstimulated). Unpaired t-test. * $p < 0.05$, **** $p < 0.0005$.

2. Fig. 2: RNA-Seq was performed with purified CD14⁺ monocytes. I did not find information how the CD14⁺ monocytes were purified and how pure they really were. Contamination by small subsets of other cells (e.g. pDCs, neutrophils, or non-classical monocytes) may have a major impact on the mRNA-seq data.

We thank the reviewer for this comment. CD14⁺ monocytes were sorted in a FACS Aria from ficoll-enriched PBMC using the following gating strategy: size and granularity-based gating on FSC vs SSC, live cells (propidium iodide), CD3⁻CD19⁻CD66b⁻CD16⁻CD56⁻CD14⁺. Post-sort purity was determined in all samples by flow cytometry (CD14 staining) and was above 98% in all samples. We have included this information in the method section of the manuscript (lines 1138-1141).

3. Fig. 3: This figure is supposed to suggest that CD14⁺ monocytes from COVID-19 patients express less TNF α , CD40 and other molecules than CD14⁺ monocytes from HV upon *in vitro* challenge with SARS-CoV-2. If I understand the methods section correctly *in vitro* stimulation was performed with total PBMC and not with purified CD14⁺ monocytes. Therefore the differences found in monocytes could be secondary to a

differential response of other cells (e.g. T cells, pDCs) to SARS-CoV-2. How would the data look like with purified CD14+ monocytes? Are the changes consistent with the mRNA-seq data of CD14+ monocytes stimulated with SARS-CoV-2?

The reviewer is correct, and the initial experiments were performed stimulating total PBMC (due to the small cell numbers available per sample). We have repeated the experiments with isolated CD14+ monocytes and the results follow the same trend as those obtained with total PBMC (Figure 3), suggesting that overall, the effects observed in monocytes are not secondary to a response of other cell types present in PBMC cultures (Revision Figure 3). We have included this information as a new Supplementary Figure (Supplementary Figure 7) and have updated the Method section accordingly (lines 1131-1133).

Revision Figure 3. Impaired *ex vivo* pathogen sensing by isolated monocytes in COVID-19. CD14+ cells were isolated from total PBMC of healthy (white bars, n=5), mild (light blue bars, n=5) and moderate (dark blue bars, n=5) COVID-19 patients by magnetic sorting and stimulated with UV-inactivated SARS-CoV-2 as in Figure 3. **a.** Summary of percentage of TNF- and IL-10-secreting monocytes in each participant group. **b.** Summary of HLA-DR, CD40, CD80 and CD86 protein expression (gMFI) in each participant group measured by flow cytometry. Data represented as mean±s.e.m. One-way ANOVA with Tukey's correction for multiple comparisons. *p<0.05, **p<0.005.

The changes observed in protein levels are also consistent with the normalized gene count values obtained from the RNA-seq data of CD14+ monocytes stimulated with UV-inactivated SARS-CoV-2 (Revision Figure 4), with a statistically significant decrease in *TNF*, *CD40* and *CD86* normalized read counts and a trend towards downregulation of *HLA-DRA* in moderate COVID-19 as compared to healthy monocytes. The only discrepancies are the expression of *IL10* and *CD80* genes. *IL10* is significantly upregulated and *CD80* is significantly downregulated in monocytes from moderate COVID-19 patients, but not at protein levels in either total PBMC or isolated monocytes. However, a trend towards increased cytokine production of IL-10 can be observed in isolated monocytes (Revision Figure 3). While the

difference is not statistically significant, it opens the possibility that IL-10 expression might be in part differentially regulated in total PBMC and isolated CD14⁺ cell cultures (Revision Figure 4).

Revision Figure 4. Normalized read counts for Figure 3 markers in CD14⁺ monocytes from healthy individuals (white bars) and moderate COVID-19 patients (blue bars) subjected to RNA-seq after UV-inactivated SARS-CoV-2 stimulation. Unpaired t-test. *p<0.05, ***p<0.001.

4. **Fig. 4:** The main conclusion drawn by the authors from this mRNA-seq of SARS-CoV-2-stimulated CD14⁺ monocytes is a pro-thrombotic signature of monocytes from COVID-19 patients. The “pro-thrombotic signature” is also mentioned in the title of the manuscript. Thus I would like to see experimental data (e.g. in vitro assays for coagulation) that CD14⁺ monocytes from COVID-19 patients are prothrombotic if exposed to SARS-CoV-2. Just the interpretation of gene signature data is not sufficient to draw this conclusion. In addition, it would be interesting to know if monocytes from COVID-19 patients with clinically proven thrombosis have a more pronounced pro-thrombotic signature or are more pro-thrombotic in in-vitro assays.

Revision Figure 5. Monocytes from moderate COVID-19 patients are functionally pro-thrombotic. Monocytes were isolated from healthy individuals, mild and moderate COVID-19 patients and left either unstimulated or stimulated with UV-inactivated SARS-CoV-2. **A.** Representative dot plots of the expression of CD41 on unstimulated (upper row) or SARS-CoV-2-stimulated (lower row) monocytes from healthy individuals (left), mild (middle) and moderate (right) COVID-19 patients (n=4 individuals per group) after co-culture with freshly isolated platelets from a healthy individual. Numbers in each dot plot represent CD41 gMFI. **b, c.** Summary of CD41 gMFI on unstimulated (**b**) or SARS-CoV-2-stimulated (**c**) monocytes after co-culture with healthy donor platelets. **d.** RNA-seq datasets from *ex vivo* isolated monocytes from moderate COVID-19 patients were grouped into low and high plasma D-dimer concentrations. Heatmap of z-score-transformed normalized read counts of significantly upregulated genes in “hemostasis” and “platelet activation, signaling and aggregation” pathways in healthy (n=6), low D-dimer concentration (n=4) and high D-dimer concentration (n=6) moderate COVID-19 monocytes. Gene expression values are scaled by column, and each row represents one individual. **e.** Summary of normalized gene counts of the genes in **d.** **f.** RNA-seq data from SARS-CoV-2-stimulated monocytes from moderate COVID-19 patients were grouped into low and high D-dimer concentrations. Heatmap of z-score-transformed normalized read counts of significantly upregulated genes in “Integrin cell surface interactions”, “Extracellular matrix organization”, “Response to elevated platelet cytosolic Ca²⁺”, “Signaling by PDGF”, “Hemostasis” and “Platelet aggregation (plug formation)” pathways in healthy (n=12), low D-dimer concentration (n=8) and high D-dimer concentration (n=6) moderate COVID-19 monocytes. Gene expression values are scaled by column, and each row represents one individual. One-way ANOVA with Tukey’s correction for multiple comparisons for **b, c, e.** *p<0.05, **p<0.005.

We thank the reviewer for this comment. In order to functionally confirm the pro-thrombotic gene expression signature of monocytes from COVID-19 we have performed *in vitro* experiments to test the capacity of monocytes from healthy individuals and COVID-19 patients to form monocyte-platelet aggregates (MPA), which are an important initiation factor in the generation of thrombi {Tutwiler, 2016, 26518435; Michelson, 2001, 11571248; Shih, 2016, 27270163} (Revision Figure 5). Isolated monocytes were co-cultured with freshly isolated platelets from a healthy individual to rule out the possibility that differential activation of platelets from healthy individuals and COVID-19 patients would lead to confounding results on the pro-thrombotic capacity of the monocytes when co-cultured. The experiments were performed with either *ex vivo* isolated monocytes (no prior stimulation) or with monocytes stimulated with UV-inactivated SARS-CoV-2 for 20 hours before platelet co-culture, following the same stimulation protocol as in the rest of experiments of this manuscript. 20 hours after setting up the co-cultures, we examined the generation of MPA by measuring the expression of CD41 in monocytes (Revision Figure 5A, 5B), which is a standard method to identify MPA (CD41 is a marker of megakaryocytes and it is not expressed on monocytes, {Hottz, 2020, 32678428}). Our results confirm that monocytes from patients with moderate COVID-19 aggregate significantly more platelets than those of healthy individuals or patients with mild COVID-19, functionally supporting the pro-thrombotic signature obtained in the RNA-seq analysis. Interestingly, the increase in MPA formation was observed after SARS-CoV-2 stimulation but also in *ex vivo* unstimulated monocytes, which was initially unexpected. However, upon detailed examination of enriched pathways in the RNA-seq analysis from *ex vivo* unstimulated monocytes, we identified a number of significantly upregulated genes in COVID-19 monocytes compared to healthy individuals that belong to pathways that include “Hemostasis” and “Platelet activation, signaling and aggregation” (Supplementary Table 4). Enrichment of none of the two pathways was statistically significant when considering their adjusted p value, but upregulation of the individual genes were, which is in agreement with the higher MPA formation capacity of *ex vivo* COVID-19 monocytes (Revision Figure 5A).

In order to address the additional question of the reviewer, i.e. to determine whether monocytes from COVID-19 patients with clinically proven thrombosis have a more pronounced pro-thrombotic signature, we took advantage of our RNA-seq datasets. As we did not have access to diagnosis of clinical thrombosis in all participants, we harnessed our RNA-seq samples, for which we had clinical information on the concentration of plasma D-dimer at the time of blood collection.

We used the plasma concentration of D-dimer to classify samples into low and high D-dimer values. D-dimer is routinely used in the clinic as a biomarker for activation of the coagulation and fibrinolysis systems {Weitz, 2017, 29096812}, and it has been extensively investigated for the diagnosis, monitoring and treatment of venous thromboembolic diseases, for which it is used routinely {Bockenstedt, 2003, 14507947}. Moreover, plasma D-dimer concentration positively correlates with venous thromboembolic diseases in the general population and in patients with COVID-19 {Berger, 2020, 32840379}.

Most patients whose monocytes were used for RNA-seq had elevated D-dimer concentrations at the time of blood collection (>500 ng/ml, Revision Table 1), but we could clearly identify two groups: those with D-dimer concentration >1600 ng/ml (High D-dimer, range 1,647-20,000 ng/ml) and those with D-dimer concentration <1,000 ng/ml (Low D-dimer, range 966-534 ng/ml).

Patient ID	D-dimer concentration (ng/ml)	Assigned group
01-016	534	Low
01-019	20000	High
01-058	966	Low
01-059	604	Low
01-066	875	Low
01-070	810	Low
01-071	3381	High
01-077	2504	High
01-080	1945	High
01-083	1647	High
01-096	635	Low
01-097	625	Low
01-098	2100	High
01-099	2921	High
01-105	751	Low
01-106	1947	High

Revision Table 1. Plasma D-dimer concentration at the time of blood collection corresponding to the moderate COVID-19 patients from whom RNA-seq samples of isolated monocytes (unstimulated and SARS-CoV-2-stimulated) were prepared.

After classifying the RNA-seq datasets into these two groups, we examined the expression of genes belonging to the pathways potentially associated with hemostasis and coagulation in both *ex vivo* RNA-seq data and RNA-seq data from SARS-CoV-2-stimulated monocytes. Normalized gene counts were z-score-converted and plotted in a heatmap in 3 groups: healthy, moderate COVID-19 with low D-dimer values and moderate COVID-19 with high D-dimer values (Revision Figure 5).

Using the *ex vivo* RNA-seq samples, we selected those genes that were significantly enriched in two pathways: “Hemostasis” and “Platelet activation, signaling and aggregation” (genes and pathways can be found in Supplementary Table 4). In agreement with the increased capacity of COVID-19 monocytes to form MPAs, we observed an increased expression of genes associated with these pathways in a D-dimer concentration-dependent manner (Revision Figure 5d). Moreover, while not all genes were significantly upregulated in the high D-dimer group, 10 out of 12 genes showed a trend toward an increased gene expression as D-dimer concentration

increased (Revision Figure 5e). This trend was specific for these two hemostasis related pathways, as it was not observed when we examined the expression of genes enriched in another significantly enriched pathway in *ex vivo* monocytes (i.e. Transmembrane transport of small molecules), with only 5 out of the 14 enriched genes showing a trend towards a D-dimer concentration-dependent increase in expression (Revision Figure 6).

a Transmembrane transport of small molecules

b

Revision Figure 6. The increased pro-thrombotic signature in high D-dimer patients is pathway specific. RNA-seq data from *ex vivo* isolated monocytes of moderate COVID-19 patients were grouped into low and high plasma D-dimer concentrations. **a.** Heatmap of z-score-transformed normalized read counts of significantly upregulated genes in “Transmembrane transport of small molecules” pathway in healthy (n=6), low D-dimer concentration (n=4) and high D-dimer concentration (n=6) moderate COVID-19 monocytes. Gene expression values are scaled by column, and each row represents one individual. **b.** Summary of normalized gene counts of the genes in **a.** One-way ANOVA with Tukey’s correction for multiple comparisons for **b.** *p<0.05, **p<0.005.

We also examined the expression (normalized read counts) of those genes enriched in pathways potentially related to a pro-thrombotic signature in RNA-samples from COVID-19 monocytes stimulated with UV-inactivated SARS-CoV-2, again dividing the samples into plasma low and high D-dimer concentrations (Revision Figure 5f). The pathways included were “Integrin cell surface interactions”, “Extracellular matrix organization”, “Response to elevated platelet cytosolic Ca²⁺”, “Signaling by PDGF”, “Hemostasis” and “Platelet aggregation (plug formation)” (Genes and pathways can be found in Supplementary Table 7, 81 genes in total). Interestingly, while all the genes were significantly upregulated in COVID-19 patients as compared to healthy controls, no apparent differences or trends in their expression were found between the two groups of patients with low and high plasma D-dimer concentration when values were plotted in a heatmap built with z-score transformed normalized gene counts of all the genes within the abovementioned pathways (Revision Figure 5f). Moreover, heatmaps of individual pathways yielded the same result (Revision Figure 7). This observation suggests that secondary SARS-CoV-2 sensing may switch monocyte phenotype and functionality in a D-dimer concentration-independent manner. We have included a new Figure (Figure 5) and two Supplementary Figures (Supplementary Figures 12 and 13) in the manuscript with these results (lines 330-401).

Revision Figure 7. Secondary SARS-CoV-2 stimulation increases the expression of pro-thrombotic genes in a D-dimer concentration independent manner. RNA-seq datasets from SARS-CoV-2-stimulated monocytes from moderate COVID-19 patients were grouped into low and high plasma D-dimer concentrations. Heatmap of z-score-transformed normalized read counts of significantly upregulated genes in “Integrin cell surface interactions”, “Extracellular matrix organization”, “Response to elevated platelet cytosolic Ca²⁺”, “Signaling by PDGF”, “Hemostasis” and “Platelet aggregation (plug formation)” pathways in healthy (n=12), low D-dimer concentration (n=8) and high D-dimer concentration (n=6) moderate COVID-19 monocytes. Gene expression values are scaled by column, and each row represents one individual.

5. Fig. 5: Again only an interpretation of gene signature data. Please show that CD14+ monocytes from COVID-19 patients are less sensitive to LPS by comparing the LPS response of monocytes from COVID-19 patients and HV.

We thank the reviewer for this comment. Figure 3 of the original manuscript shows that monocytes from COVID-19 patients are less sensitive to LPS, expressing significantly reduced levels of CD40 and TNF (Figures 3C and 3F of submitted manuscript). We have further confirmed these data, that were obtained after stimulation of total PBMC, in isolated monocytes (Revision Figure 8). We have included a Supplementary Figure in the manuscript with these data (Supplementary Figure 14).

Revision Figure 8. Impaired *ex vivo* response to LPS by isolated COVID-19 monocytes. Monocytes from healthy individuals, mild and moderate COVID-19 patients (n=4 individuals per group) were isolated by CD14 positive magnetic selection and stimulated with 100 ng/ml LPS for 20 hours. Bar graphs summarize the percentage of TNF- (left), IL-10-producing CD14⁺ cells (middle) and expression of CD40 (right) measured by flow cytometry. One-way ANOVA with Tukey’s correction for multiples comparisons. *p<0.05.

Reviewer #2 (expertise in innate immunity, monocytes and macrophages):

6. The nature of the pro-thrombotic phenotype (id est, genes supporting this conclusion) should be presented in the abstract section. This is important because most of these genes might have other functions as well.

We thank the reviewer for this comment. However, we prefer to include in the abstract only the pathways that are potentially related to thrombosis and enriched in COVID-19 monocytes instead of specific genes. The reason for this is that we have not identified a core set of genes that are predictors of the pro-thrombotic phenotype, but we have identified a set of pathways enriched in

COVID-19 monocytes that are associated to thrombi formation. We have however detailed the pathways in the abstract (lines 37-39).

7. The approach is essentially descriptive and do not provide any mechanistic on how this state of pro-thrombotic phenotype is promoted.

We thank the reviewer for this comment. We have now generated more data on the pro-thrombotic phenotype of COVID-19 monocytes (please see Reviewer 1, point 4 for details). Specifically, we have demonstrated functionally that monocytes from COVID-19 patients with moderate disease form increased monocyte-platelet aggregates compared to mild patients or healthy donors. Interestingly, this effect is not only observed upon *ex vivo* secondary stimulation of monocytes with inactivated SARS-CoV-2 as our RNA-seq data suggested, but also in *ex vivo* unstimulated monocytes from moderate COVID-19 patients. Moreover, we have included a paragraph in the discussion providing a potential mechanistic explanation of how this pro-thrombotic phenotype could be acquired (lines 443-457).

8. A important flaw in design is to restrict the analysis to CD14+ monocytes, because this population could give rise to CD16+ monocytes as well. Therefore, a more thorough assessment of all monocytes populations could have been interesting.

We respectfully disagree with the reviewer's comment that restriction to CD14⁺ classical monocytes is a flaw in the study design. The restriction to study only classical monocytes in our work is not a flaw, but the result of having examined the frequency of CD14⁺CD16⁺ and CD14^{low}CD16⁺⁺ monocytes in our cohort of available patient samples. The frequency and numbers of intermediate and non-classical monocytes were very small in our samples and did not allow for reproducible tSNE analysis nor isolation for functional experiments. Moreover, we initially tried to sort the 3 populations of monocytes to carry out bulk RNA-seq analysis and the numbers of cells obtained were too low to obtain good quality RNA for sequencing. Therefore, we decided to focus on classical monocytes. While we agree a more thorough examination of all monocyte populations (and perhaps all circulating dendritic cell populations as well) would have been ideal, we believe the results communicated in the current version of the manuscript constitute an important observation, and in its current form, additional myeloid populations is out of the scope of the paper.

9. Despite the identification of multiple epigenetic and metabolic phenotypes, no clear mechanistic frame/model is provided to explain the installation of this pro-thrombotic phenotype. As such, the study remains essentially descriptive. Interestingly, the phenotypic alterations seems more pronounced in moderate than mild COVID-19 donors.

We thank the reviewer for this comment. We have included a paragraph in the discussion with a potential mechanistic model to explain the generation of this pro-thrombotic phenotype (lines 443-457).

10. Functionally, there is no proper assessment of the alleged pro-thrombotic activity of monocytes. Any functional assessment of the pro-thrombotic activity of SARS-Cov2 stimulated monocytes would increase greatly the impact of the study.

We thank the reviewer for this comment. We have now functionally assessed the pro-thrombotic activity of monocytes. Please see Figure 5 of the revised manuscript and Reviewer 1, point 4 for additional information.

11. This sentence needs to be edited.

47. Moreover, contrasting

48. 51 observations regarding the development of cytokine storms vs. immunosuppression^{4,5} and the overactive or deficient type I IFN response in the lungs and in peripheral blood⁶⁻¹¹ have been 12 described for the role of myeloid cells in COVID-19.

We have now edited this sentence.

12. Figure 1, Lign 104 “suggest an altered activation profile skewed towards an inhibitory phenotype”. Where mixed leukocyte reaction with allogenic T cells performed to support this hypothesis?

Mixed leukocyte reactions were not carried out for the initial submission of this manuscript. However, we observed (in Figure 1) decreased expression of class II HLA and increased expression of class I HLA together with differential expression of inhibitory receptors in COVID-19 monocytes compared to healthy individuals. Therefore, we have now examined the capacity of CD14⁺ monocytes to activate SARS-CoV-2-specific T cells (Revision Figure 9). For this, magnetically isolated CD14⁺ monocytes from healthy individuals and patients with COVID-19 were loaded or not with UV-inactivated SARS-CoV-2 and co-cultured with autologous T cells (magnetically isolated CD3⁺ T cells, subsequently labelled with CFSE) at a 1:1 monocyte:T cell ratio for 10 days at 37°C, 5% CO₂. After 10 days, cultures were stained for viability, CD3, CD4, CD8 and CD14 and run on a Fortessa flow cytometer. The frequency of SARS-CoV-2-specific T cells was measured as the percentage of CFSE^{low} cells in the co-cultures. The results suggest that CD14⁺ monocytes from COVID-19 patients can efficiently activate SARS-CoV-2-specific CD8⁺ T cells, but not CD4⁺ T cells, potentially due to the alterations in class I and class II expression and other costimulatory molecules. We have included a supplementary figure with these data (Supplementary Figure 2).

Revision Figure 9. CD14⁺ monocytes from COVID-19 patients effectively activate SARS-CoV-2-specific CD8⁺ T cells, but not CD4⁺ T cells. *Ex vivo* isolated CD14⁺ monocytes from healthy individuals (white bars, n=4), mild (light blue, n=4) and moderate (dark blue, n=5) were loaded with UV-inactivated SARS-CoV-2 or vehicle and co-cultured with CFSE-labeled CD3⁺ T cells at a 1:1 monocyte:T cell ratio for 10 days. Percentage of SARS-CoV-2-specific CD8⁺ (a) and CD4⁺ (b) T cells measured by CFSE dilution. One-way ANOVA with Tukey's correction for multiple comparisons. *p<0.05, **p<0.005, ***p<0.001.

13. Figure 1. The number of samples in each clinical group should be indicated in the main text.

This suggestion has been included in the main text (lines 97, 102-103).

14. Cluster analysis: Like it is presented, this analysis is hard to read and interpret. The authors do not provide any statistical evidence than the cluster distribution change (or not) between clinical status group. The % of each cluster inside each group should be compared and the differences probed by appropriate statistical test. Otherwise, the pie chart are not informative. In addition, differences are difficult to visualize given the high number of clusters used. Rand index could be calculated to probe the differences between groups.

We thank the reviewer for this very helpful suggestion. We have now calculated the percentage of cells in each identified cluster for every individual sample grouped into healthy, mild and moderate COVID-19, to determine whether the differences in frequencies for each certain pie slice among clinical groups are statistically significant (Revision Figure 10). Interestingly, statistically significant differences in the size of clusters 1, 3, 4, 6, 8, 11, 12 and 15 were found among groups. In particular, both mild and moderate COVID-19 patients displayed reduced frequency of clusters 3 and 15, while moderate patients had a significant increase in the size of clusters 6 and 8 compared to both mild patients and healthy individuals. Finally, differences in the size of specific clusters were observed between mild and moderate COVID-19 patients. Thus, mild patients had significantly higher frequency of cells in clusters 1, 4, 11 and 12 as compared to moderate patients, and healthy individuals (except for cluster 4). We have included a supplementary figure to the manuscript (Supplementary Figure 4) with these data and have updated the results section appropriately (lines 129-136).

Revision Figure 10. Percentage of CD14⁺ monocytes from each clinical group in each Phenograph cluster. Each dot represents an individual, grouped into healthy (white), mild (light blue) and moderate (dark blue) COVID-19. The boxes extend from the 25th to the 75th percentile and the whiskers are drawn down to the minimum and up to the maximum gMFI value obtained in each group. One-way ANOVA with Tukey's correction for multiple comparisons. *p<0.05, **p<0.005, ***p<0.001, ****p<0.0005.

15. Overall assessment of H3K27Ac, H3K4me3, H3K9me2, H3K27me3, alone, is poorly informative. It should be explained/clarified that this analysis is performed at the single cell level by FACS.

We thank the reviewer for this comment. Based on this suggestion and reviewer 3's comment (point #27), we have decided to remove the epigenetic data from the revised version of the manuscript as we agree that the data presented add little information to the overall message of the manuscript.

16. Figuer 2C result is poorly reported and discussed in the text section which focuses on sup fig.

We thank the reviewer for drawing our attention to this. Figure 2c was actually described in the main text, but there was no reference to what panel was being described. We have included that information in the revised manuscript (lines 172, 177).

17. What is the metabolic prediction/biological meaning from the sphingolipid genes regulation?

We thank the reviewer for this comment. Sphingosine-1-phosphate (S1P) signaling is a complex system that has traditionally been associated with the regulation of immune cell migration. In fact, S1P directs the egress of T cells, monocytes/macrophages and other immune cells from the lymph nodes to the circulation and subsequently to the target tissue. However, there is a growing body of works suggesting that S1P metabolism is also involved in modulating immune cell functions, including cytokine secretion, activation and cytotoxic capacity. Monocytes express all 5 S1P receptors and can secrete S1P, suggesting that their migratory and functional properties could be modulated by S1P signaling. In our data, the genes significantly upregulated in COVID-19 monocytes in the sphingosine pathway (Supplementary Table 4) are all enzymes that control the generation of sphingosine and its active form, S1P, and the generation of a related sphingolipid, ceramide. Our hypothesis is that there is a sphingosine metabolism rewiring in COVID-19 monocytes that could both affect monocyte migration to target tissues and the development of innate immune functions. However, further functional experiments (enzyme activity, migration assays, gene silencing on monocytes, etc.) would need to be performed to have a clearer idea of the biological significance of the pathway in controlling monocyte migration and/or function.

18. The difference between 2C and 2h should be explained on the figure display to ease the reading.

We have included the text "upregulated genes" in Figure 2c and "downregulated genes" in Figure 2h to make the panels clearer.

19. Seahorse data should be nrought back into the main figure to support the downregulation of glycolytic activity.

We thank the reviewer for this suggestion. We have included the Seahorse data as panel k in Figure 2.

20. Figure 3. Does the poor response of mild COVID monocyte ex vivo translate into impaired T cell responsiveness (in a MLR assay)?

We thank the reviewer for this comment. Please see point 12.

21. Figure 4. The authors report the identification of a pro-thrombotic signature by analysing SARS-Cov2 stimulated (ex vivo) monocytes. Pathway analysis identifying

“hemostasis and coagulation” should be brought back into the main figure. As stated by the authors, these results are in agreement with reports of hypocoagulability in COVID patients.

We apologize if we have not understood this comment entirely, but pathway analysis identifying “Hemostasis” and others potentially involved in thrombotic events is already in the main figure (Figure 4C).

22. Functional assays assessing the pro-thrombotic activity of monocytes of ex vivo stimulated monocytes would further strengthen the message.

We thank the reviewer for this comment. We have performed functional experiments that are included in Figure 5. In addition, please see Reviewer 1, point #4.

Reviewer #3 (expertise in macrophages, COVID-19, transcriptional regulation):

23. Problematic interpretation when comparing mild and moderate COVID-19 cohorts:

There is insufficient descriptions of the 2 cohorts (mild and moderate COVID-19). Based on the information provided, the two cohorts appear poorly matched in terms of the time post symptom onset at which monocytes were collected. Previous profiling studies during viral infections (such as Dengue virus) and sepsis have showed time from symptom initiation is a strong predictor of peripheral blood responses. The average time from symptom initiation to sample collection for the mild group is almost 14 days. In immunocompetent individuals with mild disease we would expect the vast majority, if not all, of these individuals to have no recoverable virus and to be clinically well. Thus, the mild cohort is really an analysis of convalescence response to an acute COVID-19 infection? The moderate cohort samples were taken much earlier, soon after hospitalization, and at the point of worsening clinical disease. Given these monocytes are examined ex vivo, differences in monocyte responses between mild and moderate COVID-19 could reflect differences in monocytes after (mild) and during (moderate) COVID-19?

We thank the reviewer for this comment. Interestingly, when we look at the time from positive PCR to blood sample collection, both groups have more similar timings, which means that by the time of sample collection patients would potentially have detectable virus by PCR. Moreover, based on the results obtained in this manuscript and the symptoms described by mild patients at the time of sample collection (runny nose, fatigue, cough, anosmia, etc), we believe mild patients were not recovered from infection and therefore could not be considered convalescent. More information about the participants has been included in Supplementary Table 1.

24. Additional patient characteristics should also be provided: - include ranges, in addition to average, for each group as well as vaccination status (fully vaccinated, boosted, unvaccinated).

We thank the reviewer for this suggestion. We have now included ranges of age, and number of participants in each sex category in Supplementary Table 1. We have further highlighted in the methods section (“Participants”) that all participants in the study were unvaccinated (lines 1017-10180).

25. The healthy cohort has N/A listed for all comorbidities, which suggests that anyone with any preexisting conditions (hypertension, diabetes, etc.) were excluded from this group. Is that correct? As expected, the moderate group has a significant number of

comorbidities. Is it known if underlying comorbidities in the moderate group would impact the baseline expression of the monocyte markers evaluated and thus confound the analysis.

We thank the reviewer for this comment and we apologize for having used the abbreviation “N/A” indistinctively to describe two different situations. In the Mild column, “N/A” for hypertension should have been substituted for 0, as there were no participants with hypertension in this cohort. We have now corrected this. As for healthy individuals, we did not have information about most of the comorbidities listed for COVID-19 and that’s why we listed them as “N/A”. We have now obtained such information and have listed it in Supplementary Table 1. Regarding the effect of comorbidities on the baseline expression of markers examined in monocytes, to our knowledge, there are no studies that have addressed this specific question. However, some of these comorbidities, when studied as single entities, have been shown to alter the phenotype and/or frequency of peripheral blood immune cells when compared to healthy individuals, so the possibility that the reviewer raises cannot be ruled out in our study.

26. Lack of mechanism. The authors show differences in transcription, histone modifications and metabolism in monocytes, all of which have been shown following sepsis. Each of these effects could contribute to the poor response during ex vivo immune stimulation. More mechanistic insight into which mechanism is important in COVID-19 monocytes would be a great contribution that would elevate the importance of this manuscript.

We thank the reviewer for this comment. We have performed some additional experiments to confirm that monocytes from COVID-19 patients are functionally more pro-thrombotic. In addition, we have added a paragraph in the discussion section on the potential mechanisms and kinetics of events that could be leading to the dysfunctional phenotype observed in COVID-19 monocytes (lines 443-457). In summary, our hypothesis is that the initial inflammatory response observed upon infection leads to the activation of the pro-thrombotic signature which is accompanied by diminished innate immune functions as disease progresses.

27. Additional evidence is needed to completely support claims: The authors claim “monocytes displayed defects in the epigenetic remodelling” which is supported by data that shows differences in total levels of modified histones. Although changes in levels may have implications for the underlying gene regulation, it's not necessarily clear what differences will actually mean. Location/locus dependent changes (i.e. ChIP) are much better correlated to relative changes in expression and should be provided to support differences in “epigenetic remodelling”. Genome-wide ChIP-seq correlated to gene expression could provide support for claims of epigenetic remodeling as an important driver of changes in gene expression.

We thank the reviewer for this comment, that we fully agree with. Unfortunately, due to a limitation in the number of samples (and number of PBMC per sample) that we had available, we have not been able to perform ChIP-seq. Moreover, the results from those experiments would in themselves constitute an independent piece of work. Therefore, taking also into account the comment from reviewer 2 that the “assessments of the histone marks by FACS are poorly informative”, we have decided to remove these data from the manuscript. We have modified the main text accordingly.

28. The authors claim that “transcriptionally, COVID-19 monocytes switched their gene expression signature from canonical innate immune functions to a pro-thrombotic

phenotype". This claim is based on data from *ex vivo* monocytes that were further stimulated *ex vivo* (Fig 4). Given that this phenotype does not appear to be present in the analysis of *ex vivo* monocytes (Fig 2), it is not clear that this actually happens during COVID-19 infection *in vivo*.

We thank the reviewer for this interesting comment. In fact, *ex vivo* monocytes also displayed enrichment of pathways related to Hemostasis and Platelet activation and signaling, which are directly suggestive of a pro-thrombotic phenotype (Supplementary Table 4). These pathways did not reach statistical significance when adjusted p values were calculated, but the genes enriched in them were significantly upregulated in *ex vivo* monocytes from COVID-19 patients compared to healthy individuals. Therefore, in order to functionally confirm the pro-thrombotic phenotype, we have performed additional assays to determine the capacity of monocytes to form monocyte-platelet aggregates (Figure 5). Moreover, please see Reviewer 1, point #4 for additional information. Our results suggest that the increased pro-thrombotic capacity of monocytes occurs both *ex vivo* and after secondary pathogen sensing, supporting the hypothesis that this switch occurs during SARS-CoV-2 infection *in vivo*.

29. Clinically, the amount of SARS-CoV-2 circulating in the blood of moderately ill patients is relatively low (or not detectable). What concentration of inactivated virus was used to stimulate the cells *ex vivo*? Is it comparable to *in vivo*?

We thank the reviewer for this comment and we apologize for the missing information. The amount of SARS-CoV-2 utilized to stimulate the cells is now stated in the methods section (lines 1127-1128, 10^6 UV-inactivated viral particles per 10^6 cells). We also apologize for not understanding completely what the reviewer means by "is it comparable to *in vivo*?". The concentration of virus present in an infected person varies depending on many different variables, including the anatomical location being examined, the time after initial infection and even the method and sample type to measure viral concentration. Therefore it is difficult to ascertain whether the concentrations we are using are "comparable to *in vivo*" for each individual, as we do not have information of their viral load at any point and in any anatomical location (except information about a positive diagnostic PCR). However, it is worth mentioning that overall, the *ex vivo* and *in vitro* stimulations of human and mouse immune cells are models with all their advantages and limitations, and in the majority of cases, concentrations of stimuli that are above the "*in vivo*" concentrations are used (e.g. *ex vivo* MOG stimulation to identify MOG-specific T cells, LPS stimulation of antigen-presenting cells, anti-CD3 and anti-CD28 stimulation to T cell activation, virus infections, etc.). In our particular case, we used a concentration of virus that we have routinely used for other *in vitro* experiments with monocytes and various common respiratory viruses {de Marcken, 2019, 31662487}.

30. Additionally, the prothrombotic phenotype is based on the variable upregulation (Fig 4d) of a small subset of genes, many with functions outside of thrombosis (such as integrins, collagens and MMPs) (Fig 4b), in *ex vivo* monocytes stimulated with unclear quantities of inactivated virus. Based on this data, the claim of a prothrombotic expression signature is not well supported.

We thank the reviewer for this comment. We have included the concentration of virus for the experiments in the method section. Moreover, we have now performed additional experiments to functionally confirm the pro-thrombotic capacity of COVID-19 monocytes (both unstimulated and stimulated monocytes). Please see Reviewer 1, point 4 for more details on these experiments.

- 31. Figure 1a – text states “monocytes from healthy individuals were clearly distinct from both mild and moderate COVID-19 on a tSNE plot” but legend states “a. tSNE plots obtained from a concatenated sample consisting of PBMC from n=15 healthy individuals, n=15 mild and n=15 moderate COVID-19 patients.”**

We thank the reviewer for drawing out attention to this mistake. The Figure legend has now been corrected.

- 32. Is Figure 1a an analysis of classical monocytes or PBMCs? The specific cell types and markers analyzed to generate figure 1a should be explicitly stated in the text, legend or methods.**

We apologize for not making the explanation clearer. Figure 1a is an analysis of classical monocytes, that were stained as part of total PBMC samples and were gated appropriately to only limit the analysis to CD14⁺CD16⁻ monocytes. We have included this information in the method section of the manuscript.

- 33. The flow cytometry gating strategy with representative figures should be provided in supplementary materials.**

We thank the reviewer for this suggestion. We have included information about the gating strategy in the methods section and a Supplementary Figure (Supplementary Figure 15) with a representative example.

- 34. Figure 1b – show all data points – (in addition to box-whisker or violin plot).**

We have included all data points in Figure 1b.

- 35. Figure 1b – line 103 – The text suggests that PD1 is upregulated in moderate COVID-19 compared to control. However, PD1 expression in moderate is only up vs. mild COVID-19 but not vs. healthy control.**

We have now explained in detail the patterns of inhibitory marker expression and the groups that are compared to for each of them (lines 103-113).

- 36. Figure 2B, Supp Fig 3, Supp Table 3: Text states that “Interestingly, pathway enrichment identified glycolysis as the most enriched pathway in COVID-19 monocytes together with metabolism of lipids and lipoproteins.” Although metabolism of lipids and lipoproteins is in these lists, it does not appear to be the 2nd most enriched (as this sentence suggests) in either supp fig 3 or supp table 3.**

We thank the reviewer for this comment. We have clarified it in the main text (lines 165-167).

- 37. Figure 2C is not called out in the text until after 2f but could be presumably could be called out prior to 2D.**

We thank the reviewer for drawing our attention to this omission, which has now been corrected.

- 38. Figure 2e-g: Show all points (box-whisker vs violin plot) as well as representative flow plots for p-IRF3 and IκBa and pp65.**

We have included representative histograms and we have modified the figures 2e-2g to include all data points.

39. Figure 2f: uses an “expanded cohort” of COVID-19 patients. The clinical characteristics of the expanded cohorts should be described as part of supp table 1 or as an additional table. A brief description of what expanded means should be added to the text, is this just the addition of a few new individuals, or is this an entirely different cohort?

We thank the reviewer for this suggestion. The clinical characteristics of the individuals included in the expanded cohort are already included in Supplementary Table 1. We called it “expanded” because besides using the same individuals as the ones used for RNA-seq analysis, we included 1 more healthy control, 3 more moderate patients and 7 mild patients to confirm the *IFITM2* gene expression by real-time PCR. We have included information about the expanded cohort in the main text (lines 186-188).

40. Figure 2J and 2K: identify which cohort of individuals (initial, expanded or new) was utilized for these experiments and provide appropriate clinical details if not already done. Please show individual data points in addition to bar plots.

Experiments in Figure 2J and 2K (Figures 2K and 2L in the revised version of the manuscript) were performed with samples from the initial cohort of participants. Clinical information is already included in Supplementary Table 1. We have modified the bar plots to include all the individual data points.

41. Figure 3. The text should specifically state that these experiments are performed with inactivated SARS-CoV-2 and hCoVs.

We have included “inactivated” in the description of the results in Figure 3.

42. Figure 3. add information to text or legend regarding the duration and magnitude (what quantity of virus/stimuli (LPS)) for each stimulation.

We have included the information the reviewer requests in the main text (lines 233-234, 239-240).

43. Figure 4. state these experiments are performed with inactivated SARS-CoV-2.

This information is now included in the main text.

44. Figure 4. add information to the text or legend regarding the duration and magnitude of inactive SARS-CoV-2 stimulation.

This information has now been included in the main text.

45. Figure 4a: Figure 2a shows that baseline gene expression is different in monocytes from COVID-19 and healthy controls before they are stimulated. How much of the variance observed is due to baseline differences rather than differences in the response to inactivated SARS-CoV-2 stimulation.

We thank the reviewer for this interesting comment. In order to answer this question, we would have needed to have paired RNA-seq samples from the same individuals with and without

stimulation. Unfortunately, we only had paired samples from 3 individuals in our cohort and therefore, this question cannot be answered with the information we have generated.

46. Figure 4b: adding information to the volcano plot to show if movement to the left (-logFC) or right (+logFC) is representative of higher expression in control vs COVID would make the figure easier to interpret.

We thank the reviewer for this suggestion, that we have incorporated into Figure 4b.

47. Supp Fig 8 and Supp Table 6. Please clarify what 2 groups are being compared (inactive SARS-CoV-2 stimulated COVID-19 vs. health monocytes?) and what differentially regulated genes are used to generate these fig/tables. Is this both up and down regulated genes?

Supplementary Figure 8 and Supplementary Table 6 are created using all differentially expressed genes (upregulated and downregulated) in UV-inactivated SARS-CoV-2-stimulated monocytes from moderate COVID-19 patients compared to UV-inactivated SARS-CoV-2-stimulated monocytes from healthy controls. Differentially expressed genes were those with ≥ 1.5 fold change in COVID-19 vs. healthy monocytes and $FDR < 0.05$. Pathway enrichment in Supplementary Figure 8 was carried out with these differentially expressed genes (both upregulated and downregulated), which are listed in Supplementary Table 6.

48. Line 305 (in reference to Figure 4E). The text describes these genes as downregulated. Are they actually downregulated when compared to unstimulated? Or are they induced upon stimulation but to a lower degree than in healthy stimulated monocytes? Please adjust the text to make this distinction more clear.

We thank the reviewer for this suggestion that we have clarified in the text (line xx). Briefly, in order to properly determine whether these pathways are downregulated when compared to unstimulated samples, we would need to have paired unstimulated-stimulated samples for all the individuals used in the RNA-seq experiment. However, this was not the case due to the small amount of initial PBMC that we had per participant. Therefore, the downregulation of pathways that we refer to in the text is when comparing stimulated COVID-19 monocytes with stimulated healthy control monocytes.

49. Figure 5a and 5b: characteristics of the sepsis cohort should be described (timing of when samples were drawn, type of infection, etc.) to provide important context for the reader.

We have now included this information on the main text (lines 412-414).

50. Figure 5d – why use SARS-CoV-2 stimulation to compare with a tolerance signature that was specifically determined following LPS stimulation?

We thank the reviewer for this comment. The idea behind this comparison was to determine whether the unresponsive phenotype observed in monocytes upon SARS-CoV-2 stimulation (decreased expression of pathways that are involved in canonical innate immune function, including pro-inflammatory cytokine secretion, type I IFN response, etc.) held any similarity to the phenotype that monocytes acquire during endotoxin tolerance induction, which is also characterized by decreased expression of genes involved in canonical innate immune functions such as pro-inflammatory cytokine expression.

51. Text lines 188-194. This is a long, complex sentence that could be broken into parts to enhance reader understanding. Given that severe COVID-19 has been associated with too much inflammatory immunopathology, it would be helpful to explain why lack of upregulation of cytokine expression important? This could be done in the discussion.

We thank the reviewer for this suggestion. We have re-written the text in lines 188-194 and we have included a sentence in the discussion addressing the importance of monocyte unresponsiveness upon secondary stimulation (lines 456-459).

52. Line 842 – mild of moderate should be mild of moderate.

We have corrected the typo in the revised version of the manuscript.

53. Please provide details of how PBMCs were frozen and thawed in the methods as these could affect RNA expression.

We have included this information in the section “PBMC isolation, storage and thawing” (lines 1048-1055).

54. Each supplementary table should contain a title with clear information describing what is contained within that table. If it includes a differential expression list, then it should state what it is and what 2 groups are being compared to determine that list as well as the FC and FDR cut-off values. For example, the current titles “Table S7 - stim up pathway” is difficult to interpret. Does this mean what is up in stimulated vs unstimulated or is it moderate stimulated vs. healthy stimulated?

We thank the reviewer for this suggestion. The first tab of the Excel file, called “Legend”, describes in detail what each Supplementary Table contains. Moreover, we have included a title for the corresponding table in each of the tabs.

REVIEWERS' COMMENTS

Reviewer #1 (expertise in monocyte biology, inflammation):

The authors have addressed all of my comments in great detail and added a lot of new important data.

The manuscript is very convincing in my eyes now.

Congratulation to the nice work.

Reviewer #2 (expertise in innate immunity, monocytes, macrophages):

The authors have convincingly addressed my comments.

Specifically, and regarding my previous questions:

- The pro-thrombotic activity is now experimentally assessed providing a functional substratum to the transcriptional findings (Fig. 5).

- The T cell stimulatory activity has been assessed using a CTV dilution assay performed with autologous T cells as a proxy for antigen-specific T cells. The new sup fig. CD14+ monocytes from COVID-19 patients can efficiently activate SARS-CoV-2-specific CD8+ T cells, but not CD4+ T cells (Rev fig. 9).

- T-SNE analysis has been refined to probe for stastical differences. New findings substantiate that both mild and moderate COVID-19 patients displayed reduced frequency of clusters 3 and 15, while moderate patients had a significant increase in the size of clusters 6 and 8 compared to both mild patients and healthy individuals.

- Epigenetic analysis has been removed.

In sum, the ms is now improved in many ways.

Reviewer #3 (expertise in macrophages, COVID-19, transcriptional regulation):

Maher et al. have made significant improvements in the revised manuscript and have addressed all of my concerns.